# Towards Size-Independent Generalization Bounds for Deep Operator Nets

**Pulkit Gopalani**[*]  *gopalani@umich.edu*
*Department of Computer Science & Engineering*
*University of Michigan*

**Sayar Karmakar**  *sayarkarmakar@ufl.edu*
*Department of Statistics*
*University of Florida*

**Dibyakanti Kumar**  *dibyakanti.kumar@postgrad.manchester.ac.uk*
*Department of Computer Science*
*The University of Manchester*

**Anirbit Mukherjee**  *anirbit.mukherjee@manchester.ac.uk*
*Department of Computer Science*
*The University of Manchester*

**Reviewed on OpenReview:** *https://openreview.net/forum?id=21kOOu6LN0&noteId=21kOOu6LN0*

## Abstract

In recent times machine learning methods have made significant advances in becoming a useful tool for analyzing physical systems. A particularly active area in this theme has been "physics-informed machine learning" which focuses on using neural nets for numerically solving differential equations. In this work, we aim to advance the theory of measuring out-of-sample error while training DeepONets – which is among the most versatile ways to solve P.D.E systems in one-shot. Firstly, for a class of DeepONets, we prove a bound on their Rademacher complexity which does not explicitly scale with the width of the nets involved. Secondly, we use this to show how the Huber loss can be chosen so that for these DeepONet classes generalization error bounds can be obtained that have no explicit dependence on the size of the nets. The effective capacity measure for DeepONets that we thus derive is also shown to correlate with the behavior of generalization error in experiments.

## 1 Introduction

Deep learning has recently emerged as a competitive way to solve partial differential equations (P.D.Es) numerically. We note that the idea of using nets to solve P.D.Es dates back many decades Lagaris et al. (1998) Broomhead & Lowe (1988). In recent times this idea has gained significant momentum and "AI for Science" Karniadakis et al. (2021) has emerged as a distinctive direction of research. Some of the methods at play for solving P.D.Es neurally E et al. (2021) are the Physics Informed Neural Networks (PINNs) paradigm Raissi et al. (2019) Lawal et al. (2022), "Deep Ritz Method" (DRM) Yu et al. (2018), "Deep Galerkin Method" (DGM) Sirignano & Spiliopoulos (2018) and many further variations that have been developed of these ideas, Kaiser et al. (2021); Erichson et al. (2019); Wandel et al. (2021); Li et al. (2022); Salvi et al. (2022).

These different data-driven methods of solving the P.D.Es can broadly be classified into two kinds, **(1)** ones which train a single neural net to solve a specific P.D.E. and **(2)** operator methods – which train multiple nets in tandem to be able to solve in "one shot" a family of differential equations, with a fixed differential

---

[*]This work was done when the author was at the Department of Electrical Engineering, Indian Institute of Technology Kanpur (IIT Kanpur), India.

operator and different "source / forcing" functions. The operator net was initiated in Chen & Chen (1995) and its most popular deep net version, the DeepONet, was introduced in Lu et al. (2021). An unsupervised form of this idea was introduced in Wang et al. (2021). We note that the operator methods are particularly useful in the supervised setup when the underlying physics is not known – as is the setup in this work – and state-of-the-art approaches of this type, can be seen in works like Raonić et al. (2023).

As an explicit example of using DeepONet in the supervised setup, consider solving a pendulum O.D.E., $\frac{\mathrm{d}(y,v)}{\mathrm{d}t} = (v, -k \cdot \sin(y) + f(t)) \in \mathbb{R}^2$ for different forcing functions $f(t)$. Then using sample measurements of valid $(f, y)$ tuples, a DeepONet setup can be trained only once, and then repeatedly be used for inference on new forcing functions $f$ to estimate their corresponding solutions $y$. This approach is fundamentally unlike traditional numerical methods where one needs to run the optimization algorithm afresh for every new source function. In a recent study Lu et al. (2022), the authors showed how the DeepONet setup – that we focus on in this work – has significant advantages over other competing neural operators, like the FNO Li et al. (2020), in solving various differential equations of popular industrial use.

As a testament to the foundational nature of the idea, we note that over the last couple of years, several variants of DeepONets have been introduced, like, Tripura & Chakraborty (2023); Goswami et al. (2022a); Zhang et al. (2022); Hadorn (2022-03-16); Park et al. (2023); de Sousa Almeida et al. (2023); Tan & Chen (2022); Xu et al. (2023); Lin et al. (2023). Different implementations of neural operators have been demonstrated to be useful for various scientific applications, like for predicting crack shapes in brittle materials Goswami et al. (2022b), for fluid flow in porous materials Choubineh et al. (2023), for simulating plasmas Gopakumar & Samaddar (2020), for seismic wave propagation Lehmann et al. (2023), for weather modeling Kurth et al. (2023) etc.

In light of this burgeoning number of applications, we posit that it becomes critical that the out-of-sample performance of neural operators be mathematically understood. Towards this goal, in this work, we prove that the generalization error of one of the most versatile forms of operator learning i.e DeepONets, can become independent of the number of training parameters. We note that this is a significant improvement over the best known generalization bounds for DeepONets, Theorem 5.3 in Lanthaler et al. (2022), which grow exponentially in the number of parameters. We are able to obtain this insight via analyzing the Rademacher complexity of DeepONets. That we can recover here the usual neural net like generalization bound of the form $\frac{\text{a capacity measure}}{\sqrt{\text{sample-size}}}$ is particularly interesting because in here the predictor, the DeepONet, is a non-Lipschitz function, while nets are Lipschitz functions and that is critical to how the standard analysis is carried out for obtaining generalization error bounds.

**Motivations for Understanding Rademacher Complexity** We recall that typically there is a vast lack of information about the distribution of the obtained models in any stochastic training setup. Corresponding to a stochastic training algorithm, say ALG, that is using a $m-$ sized dataset $\mathcal{S}_m$ to search over a class of predictors $\mathcal{H}$, there is scarce mathematical control possible over the distribution of its output random variable – the random predictor thus obtained, say ALG$(\mathcal{S}_m, \mathcal{H})$. Hence there is an almost insurmountable challenge in being able to control the quantity we would ideally like to bound, the out-of-sample risk of the obtained predictor i.e. $R(\ell \circ \text{ALG}(\mathcal{S}_m, \mathcal{H}))$ - where $R$ denotes the expectation w.r.t the true data distribution, say $\mathcal{D}$, of a loss $\ell$ evaluated on the predictor ALG$(\mathcal{S}_m, \mathcal{H})$.

The fundamental idea that takes us beyond this conundrum is to relax our goals from aiming to control the above to aiming to control only the data averaged worst (over all possible predictors) "generalization gap" between the population and the empirical risk i.e, $\mathbb{E}_{\mathcal{S}_m \sim \mathcal{D}^m} \left[ \sup_{h \in \mathcal{H}} \left( \hat{R}_m(\ell \circ h) - R(\ell \circ h) \right) \right]$ – where corresponding to a choice of loss $\ell$, $\hat{R}_m(\ell \circ h)$ is the empirical estimate over the data sample $\mathcal{S}_m$ of the risk of $h$ and $R(\ell \circ h)$ is the population risk of $h$ over the data distribution $\mathcal{D}$. The importance of the statistical quantity, Rademacher complexity Bartlett & Mendelson (2001) of the function class $\ell \circ \mathcal{H}$ i.e. $\mathbb{E}_{z_1, \ldots, z_m \sim \mathcal{D}^m} \left( \mathbb{E}_\epsilon \left[ \sup_{h \in \mathcal{H}} \frac{1}{m} \sum_{i=1}^m \epsilon_i \ell \circ h(z_i) \right] \right)$ (where $\epsilon_i \sim \pm 1$ are Bernoulli random variables) stems from the fact that it is what bounds this aforementioned measure of the generalization gap.

It is to be noted that trying to control the above frees us from the specifics of any particular M.L. training algorithm being used (of which there is a myriad of heuristically good options) to find $\text{arginf}_{h \in \mathcal{H}} R(\ell \circ h)$. But on the other hand, by analyzing the Rademacher complexity we gain insight into how the choice of the

hypothesis class $\mathcal{H}$, the choice of the loss $\ell$, and the data distribution $\mathcal{D}$ interact to determine the ability to find $\mathrm{arginf}_{h\in\mathcal{H}} R(\ell\circ h)$ via empirical estimates. Suppose that there exists some constant $\mathcal{C}(\mathcal{H},\mathcal{D})$ — a capacity measure — such that $\mathbb{E}_{\mathcal{S}_m\sim\mathcal{D}^m}\left[\sup_{h\in\mathcal{H}}\left(\hat{R}_m(\ell\circ h) - R(\ell\circ h)\right)\right]$ is upper-bounded by $\frac{2\cdot\mathcal{C}(\mathcal{H},\mathcal{D})}{\sqrt{n}}$, then one can conclude that while training with $n \geq \left(\frac{2\cdot\mathcal{C}(\mathcal{H},\mathcal{D})}{\epsilon}\right)^2$ samples the data-averaged worst-case generalization error for the predictor obtained is at most $\epsilon$, for any arbitrarily small $\epsilon > 0$. This motivates one of the uses of Rademacher complexity i.e to provide such estimates on the number of samples required to achieve a target test accuracy. In particular, in cases where this capacity does not scale with the number of parameters in the hypothesis class $\mathcal{H}$, one may begin to explain why certain over-parameterized models might be good spaces for the learning task. Such understanding becomes immensely helpful when $\mathcal{H}$ is very complicated, as is the focus here, that of $\mathcal{H}$ being made of DeepONets. In Section 2, we review some of the recent advances in the Rademacher analysis of neural networks.

We crucially note (Theorem 4.13, Ma (2021)) that any condition on $m, \ell, \mathcal{H}$ and $\mathcal{D}$ that makes Rademacher complexity small is a condition which when true it becomes reasonable to expect that the empirical and the population risk could also be close for any predictor in the class $\mathcal{H}$.

In this work, we will compute the Rademacher complexity of appropriate classes of DeepONets and use it to give the first-of-its-kind bound on their generalization error which does not explicitly scale with the number of parameters. Generalization bounds that do not scale with the size of the nets can be seen as a step towards explaining the success of overparameterized architectures for that learning task. Further, our experiments will demonstrate that the complexity measures of DeepONets as found by our Rademacher analysis indeed correlate to the true generalization gap, over varying sizes of the training data.

## 1.1 Overview of Training DeepONets & Our Main Results

Following Lu et al. (2021), we refer to the schematic in Fig.1 below for the DeepONet architecture,

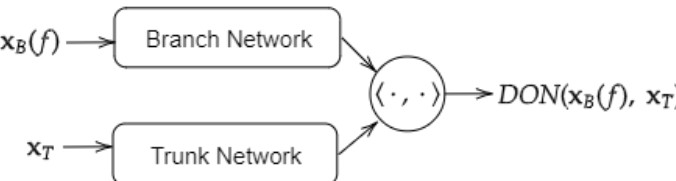

Figure 1: A Sketch of the DeepONet ( "DON") Architecture

In the above diagram, the Branch Network and the Trunk Network are neural nets with a common output dimension. $\boldsymbol{x}_B(f) \in \mathbb{R}^{d_1}$, the input to the branch net is an "encoded" version of a function $f$ i.e. a discretization of $f$ onto a $d_1$−sized grid of "sensor points" in its input domain. $\boldsymbol{x}_T \in \mathbb{R}^{d_2}$ is the trunk input. If the activation is $\sigma$ at all layers and the branch net and the trunk net's layers are named $\boldsymbol{B}_k, k = 1, 2, \ldots, q_B$ and $\boldsymbol{T}_k, k = 1, 2, \ldots, q_T$ respectively, then the above architecture implements the map,

$$\mathbb{R}^{d_1} \times \mathbb{R}^{d_2} \ni (\boldsymbol{x}_B(f), \boldsymbol{x}_T) \mapsto \mathrm{DeepONet}(\boldsymbol{x}_B(f), \boldsymbol{x}_T) :=$$

$$\left(\underbrace{\boldsymbol{B}_{q_B}(\sigma(\boldsymbol{B}_{q_B-1}(\ldots\sigma(\boldsymbol{B}_1(\boldsymbol{x}_B(f)))\ldots)))}_{=\boldsymbol{f}_B(\boldsymbol{x}_B(f))}\right)^{\top}\left(\underbrace{\boldsymbol{T}_{q_T}(\sigma(\boldsymbol{T}_{q_T-1}(\ldots\sigma(\boldsymbol{T}_1(\boldsymbol{x}_T))\ldots)))}_{=\boldsymbol{f}_T(\boldsymbol{x}_T)}\right) \quad (1)$$

In above the number of rows of the matrix $\boldsymbol{B}_{q_B}$ and $\boldsymbol{T}_{q_T}$ need to be the same for the inner-product to be defined. For a concrete example of using the above architecture, consider the task of solving the pendulum O.D.E from the previous section, $\frac{\mathrm{d}(y,v)}{\mathrm{d}t} = (v, -k\cdot\sin(y) + f(t)) \in \mathbb{R}^2$. For a fixed initial condition, here the training/test data sets would be 3−tuples of the form, $(\boldsymbol{x}_B(f), x_T, y)$ where $y \in \mathbb{R}$ is the angular position of the pendulum at time $t = \tau$ for the forcing function $f$. Typically $y$ is a standard O.D.E. solver's approximate

solution. Given $m$ such training data samples, the $\ell_2$ empirical loss would be,

$$\mathcal{L} \coloneqq \frac{1}{2m} \sum_{i=1}^{m} \left( y_i - \mathrm{DeepONet}(\boldsymbol{x}_B(f_i), \tau_i) \right)^2 = \frac{1}{2m} \sum_{i=1}^{m} \left( \mathcal{G}(f_i, \tau_i) - \mathrm{DeepONet}(\boldsymbol{x}_B(f_i), \tau_i) \right)^2 . \tag{2}$$

In above $\mathcal{G}$ is the solution operator for this O.D.E. that the O.D.E. solver can be imagined to be simulating. The rationale for this loss function above originates from the universal approximation property of DeepONets which we have reviewed as Theorem C.1. Going beyond such approximation theorems, we state the results that we prove here about the risk bounds in this novel learning setup.

**Theorem** (Informal Statement of Theorem 4.1). Consider a class of DeepONets, with absolute value activation, whose branch and trunk networks are both of depth $n$ (i.e. $q_B = q_T = n$ in equation 1) and suppose that the squared norms of the inputs to them are bounded in expectation by $M_{x,B}$ and $M_{x,T}$ respectively. Then the average Rademacher complexity, for training with samples of size $m$, is bounded by,

$$\mathcal{O}\left( \frac{2^{n-1} \mathcal{C}_{n,n-1}}{\sqrt{m}} \left( \prod_{j=2}^{n-1} \mathcal{C}_{-j,-j} \right) M_{x,B} M_{x,T} \right) \tag{3}$$

where the constants $\mathcal{C}_{n,n-1}$ and $\mathcal{C}_{-k,-k}$, $k = 2, 3, \ldots, n-1$ are defined so that the weight matrices of the Branch Network and the Trunk Network of all the DeepONets in the class satisfy the following bounds,

$$\sum_{k_1=1}^{b_{-1}} \sum_{k_2=1}^{t_{-1}} \left| \left[ \sum_{j=1}^{p} (\boldsymbol{B}_{n,j} \boldsymbol{T}_{n,j}^{\top}) \right]_{k_1, k_2} \right| \cdot \|\boldsymbol{B}_{n-1,k_1}\| \cdot \|\boldsymbol{T}_{n-1,k_2}\| = \mathcal{C}_{n,n-1},$$

$$\sup_{(\boldsymbol{v},\boldsymbol{w})} \sum_{j_1=1}^{b_{-k}} \sum_{j_2=1}^{t_{-k}} \left| (\boldsymbol{v}\boldsymbol{w}^{\top})_{j_1,j_2} \right| \cdot \|\boldsymbol{B}_{n-k,j_1}\| \|\boldsymbol{T}_{n-k,j_2}\| = \mathcal{C}_{-k,-k} \qquad \forall k = 2, \ldots, n-1, \tag{4}$$

where in each sum, above $\boldsymbol{v}, \boldsymbol{w}$ are on the unit spheres of the same dimensionality as the number of rows in $\boldsymbol{B}_{n-k}$ and $\boldsymbol{T}_{n-k}$ respectively and $b_{-k}, t_{-k}$ represent the number of rows in the weight matrices $\boldsymbol{B}_{n-k}$ and $\boldsymbol{T}_{n-k}$ respectively. (For any branch weight matrix say $\boldsymbol{B}_p$, in above we have denoted its $j$-th row as $\boldsymbol{B}_{p,j}$ and similarly for the trunk.)

Towards making the above measures computationally more tractable, in Appendix J we have shown that $\forall k = 2, \ldots, n-1$ one can choose an upperbound $\tilde{\mathcal{C}}_{-k,-k}$ in place of $\mathcal{C}_{-k,-k}$ where,

$$\tilde{\mathcal{C}}_{-k,-k} = \|\boldsymbol{X}\| \text{ for } \boldsymbol{X} \in \mathbb{R}^{b_{-k} \times t_{-k}} \text{ with } \boldsymbol{X}_{j_1,j_2} \coloneqq \|\boldsymbol{B}_{n-k,j_1}\| \cdot \|\boldsymbol{T}_{n-k,j_2}\| \tag{5}$$

In Section 4.3, we undertake an empirical study on a particular component of the generalization bound, represented as $\frac{\mathcal{C}_{n,n-1}}{\sqrt{m}}, \left( \prod_{j=2}^{n-1} \tilde{\mathcal{C}}_{-j,-j} \right)$, in relation to the generalization gap to assess the correlation between these factors.

Having proven the key theorem above, the following generalization bound with a modified loss function for DeepONets follows via standard arguments about Rademacher complexity,

**Theorem** (Informal Statement of Theorem 4.2). Considering the same class of DeepONets as in Theorem and using the Huber loss $\ell_{H,\delta}(x) \coloneqq \begin{cases} \frac{1}{2}x^2 & \text{for } |x| \leq \delta \\ \delta \cdot (|x| - \frac{1}{2}\delta) & \text{for } |x| > \delta \end{cases}$ with $\delta = \left(\frac{1}{2}\right)^{n-1}$ as the loss function. The expectation over data of the supremum of the generalization error over the above class of DeepONets can then be bounded by,

$$\mathcal{O}\left( \frac{\mathcal{C}_{n,n-1}}{\sqrt{m}} \left( \prod_{j=2}^{n-1} \mathcal{C}_{-j,-j} \right) M_{x,B} M_{x,T} \right)$$

where $\mathcal{C}_{n,n-1}$ and $\mathcal{C}_{-j,-j}$ are the constants defined in Theorem .

To the best of our knowledge, there is no general principle which determines when the worst case generalization gap of a learning scenario scales like $\frac{\mathcal{C}}{\sqrt{m}}$ for some "capacity" function $\mathcal{C}$ which is entirely determined by the data distribution and the norms of the parameters allowed in the predictor class. We recall that this

is the familiar form of this bound from calculations done with neural nets, as reviewed in Appendix 2. We posit that the reappearance of this form of the bound for DeepONets could not have been taken for granted without the involved calculation as done here to prove the above theorem.

*Secondly,* we note that there is no explicit dependence on the widths of the DeepONet of the effective capacity measure of $\mathcal{C}$ obtained above, once the depth has been fixed. In line with how terminology was developed for standard neural nets, as reviewed in Section 2, we can say that thus we have obtained size-independent generalization bounds - even for DeepONets and also while not making any assumptions on the target operator $\mathcal{G}$ - an instance of which was seen in equation 2.

*Lastly,* we note that in Pestourie et al. (2023), the authors had recently demonstrated empirically the effectiveness of using Huber loss for training neural networks to solve P.D.Es. Albeit for a different setup, that of solving P.D.E. systems, via the above theorem, we provide a theoretical justification for why Huber losses could be well suited for such tasks, and in here we are also able to derive a good value of $\delta$ –while this hyperparameter choice in Pestourie et al. (2023) was not grounded in any theory.

In Appendix B we give an experimental study of how lowering the value of $\delta$ indeed helps lower both the generalization error as well as the test error. Further, we will give experimental evidence (Figures 2 and 6) that the capacity function derived above indeed has high correlation with the variations in the generalization error over different experiments at different training data size.

In light of the above, we note the following *practical applicability* and *qualitative significance* of our bounds on the Rademacher complexity and generalization error of DeepONets.

**Choosing the Loss Function** Theorem 4.2 suggests that training on the Huber loss function could lead to better generalization. We give a performance demonstration in Appendix A for Huber loss function at $\delta = \frac{1}{2^{n-1}}$ for DeepONets using tanh activation (although our above theorem does not capture this choice of activation) and we note that it gives 2 orders of magnitude lower test error than for the same experiment using $\ell_2$ loss.

Additionally, note that, DeepONets without biases and with positive homogeneous activation, are invariant to the scaling of the branch net's layers, $\mathbf{B}_k$,and trunk net's layers, $\mathbf{T}_k$, by any $\mu_k$ and $\lambda_k > 0$ respectively $\forall k$ s.t $\prod_k (\mu_k \cdot \lambda_k) = 1$. *This is a larger symmetry than for usual nets but our complexity measure mentioned above is also invariant under this combined scaling.* This can be seen as being strongly suggestive of our result being a step in the right direction.

**Explaining Overparameterization** Since our generalization error bound has no explicit dependence on the width or the depth (or the number of parameters), it constitutes a step towards explaining the benefits of overparameterization in this new setup.

## 2 Related Works

Over the past few years, many novel generalization bounds for standard neural nets have been established. Many of these works have computed bounds on Rademacher complexity for various classes of nets, to show how different norm combinations of the involved weight matrices affect generalization performance, Sellke (2024), Golowich et al. (2018), Bartlett et al. (2017), Neyshabur et al. (2015). For shallow neural nets such methods have also been useful in explaining the benefits of dropout Arora et al. (2021) and overparameterization Neyshabur et al. (2019). However, for state-of-the-art neural nets, other approaches Dziugaite & Roy (2017), Arora et al. (2018), Neyshabur et al. (2018), Mukherjee (2021) have sometimes given tighter bounds while using bounding expressions that are computationally very expensive. A thorough discussion of the reach and limitations of these various bounds can be found in Nagarajan (2021).

In Kumar & Mukherjee (2024), the authors investigate the behavior of the $\ell_2$-distance between the true solution and the PINN surrogate, building on the work of Mishra & Molinaro (2023). While the study establishes an upper bound on the $\ell_2$-distance, this result alone is insufficient to guarantee successful training. For training to be effective, it is also necessary to demonstrate that the empirical risk converges to the population risk, which can be ensured using Rademacher complexity-based bounds.

## 2.1 Comparison to Lanthaler et al. (2022)

*Firstly,* we note that none of the theorems presented here depend on the results in Lanthaler et al. (2022). The proof strategies are entirely independent. *Secondly,* to the best of our knowledge, Theorem 5.3 in Lanthaler et al. (2022) is the only existing result on generalization error bounds for DeepONets. However, their bound has an explicit dependence on the total number of parameters (the parameter $d_\theta$ there) in the DeepONet. Such a bound is not expected to explain the benefits of overparameterization, which is one of the key features of modern deep learning (Dar et al., 2021; Yang et al., 2020).

We note that for usual implementations for DeepONets, where depths are typically small and the layers are wide, for a class of DeepONets at any fixed value of our complexity measure i.e. $\mathcal{C}_{n,n-1}\left(\prod_{j=2}^{n-1}\mathcal{C}_{-j,-j}\right)$, a generalization error bound based on our Rademacher complexity bound in Theorem 4.1 will be smaller than the one in Lanthaler et al. (2022) that scales with the total number of parameters.

Further, as pointed out in our second main result, Theorem 4.2, our Rademacher bound lends itself to an entirely size-independent generalization bound when the loss is chosen as a certain Huber loss.

**Notation** Given any $U \subset \mathbb{R}^n$, denote as $L^2(U)$ the set of all functions $f : U \to \mathbb{R}$ s.t $\int_U f^2(\boldsymbol{x})\,\mathrm{d}\mu(\boldsymbol{x}) < \infty$ where $\mu$ is the standard Lebesgue measure on $\mathbb{R}^n$. And we denote as $C(U)$, the set of all real valued continuous functions on $U$. The unit sphere in $\mathbb{R}^k$ is denoted by $S^{k-1} \coloneqq \left\{\boldsymbol{x} \in \mathbb{R}^k \mid \|x\|_2 = 1\right\}$. For any matrix $\mathbf{A}$, $\|\mathbf{A}\| = \sup_{\mathbf{v}\neq 0} \frac{\|\mathbf{A}\mathbf{v}\|_2}{\|\mathbf{v}\|_2}$ denotes its spectral norm. For any bounded set $S \subset \mathbb{R}^n$, $U(S)$ denotes the uniform distribution over that set $S$.

## 3 The Mathematical Setup

Firstly, we recall the definition of the general DeepONet architecture from Lu et al. (2021), which will be our focus in this work.

**Definition 1 (A DeepONet (Version 1)).** We continue in the setup specified in equation 1. Further, let $p$ be the common output dimension of $q_B$ depth "branch net" $\boldsymbol{f}_B$ and the $q_T$ depth "trunk net" $\boldsymbol{f}_T$.

When required to emphasize some constraint on the weights of a DeepONet, we will denote DeepONet$(\boldsymbol{x}_B, \boldsymbol{x}_T)$ by DeepONet$_{\boldsymbol{w}}(\boldsymbol{x}_B, \boldsymbol{x}_T)$, where $\boldsymbol{w}$ collectively stands for all the weight matrices in branch and trunk networks.

Corresponding to the DeepONets defined above we define the following width parameters for the different weight matrices.

**Definition 2. (Width Parameters for DeepONets)**

- For $k = 1, \ldots, q_B - 1$ and $\ell = 1, \ldots, q_T - 1$, define $b_{-k}$ and $t_{-\ell}$ to be the number of rows in the weight matrices $\boldsymbol{B}_{q_B - k}$ and $\boldsymbol{T}_{q_T - \ell}$ respectively.

- In the setup of Definition 1, we define the functions $\boldsymbol{f}'_B$ and $\boldsymbol{f}'_T$ s.t given any $m$ inputs to a DeepONet as, $\{(\boldsymbol{x}_{B,i}, \boldsymbol{x}_{T,i}) \mid i = 1, \ldots, m\}$ we have the following equalities,

$$\boldsymbol{f}_B(\boldsymbol{x}_{B,i}) = \boldsymbol{B}_{q_B}\sigma_1\left(\boldsymbol{B}_{q_B-1}\boldsymbol{f}'_B(\boldsymbol{x}_{B,i})\right), \ \ \boldsymbol{f}_T(\boldsymbol{x}_{T,i}) = \boldsymbol{T}_{q_T}\sigma_2\left(\boldsymbol{T}_{q_T-1}\boldsymbol{f}'_T(\boldsymbol{x}_{T,i})\right). \tag{6}$$

Next, we define a certain smoothness condition that we need for the activation functions used.

**Assumption 1.** Let $\phi_P, \phi_Q : \mathbb{R} \to \mathbb{R}$ be the activation functions for the branch net and the trunk net such that $\exists L > 0$ s.t for any two sets of functions valued in $\mathbb{R}$, say $\mathcal{P}$ and $\mathcal{Q}$, $\forall (p,q), (p',q') \in \mathcal{P} \times \mathcal{Q}$ and $\forall (\boldsymbol{x}, \boldsymbol{y}) \in \text{Domain}(\mathcal{P}) \times \text{Domain}(\mathcal{Q})$, the following inequality holds,

$$|\phi_P(p(\boldsymbol{x}))\phi_Q(q(\boldsymbol{y})) - \phi_P(p'(\boldsymbol{x}))\phi_Q(q'(\boldsymbol{y}))| \le L|p(\boldsymbol{x})q(\boldsymbol{y}) - p'(\boldsymbol{x})q'(\boldsymbol{y})|. \tag{7}$$

Our main results work under the above assumption and hence in particular, our results apply to the absolute value map, $\mathbb{R} \ni x \mapsto |x| \in \mathbb{R}$, being the activation function in both the branch and the trunk net. In Appendix I we will indicate why this is sufficient to also capture DeepONets with ReLU activations and being trained on bounded data.

Towards formalizing the setup of the loss functions and the training data for training DeepONets we recall the definition of the Huber loss Huber (1964).

**Definition 3** (Huber Loss). For some $\delta \geq 0$ the Huber loss is defined as

$$\ell_{H,\delta}(x) := \begin{cases} \frac{1}{2}x^2 & \text{for } |x| \leq \delta \\ \delta \cdot (|x| - \frac{1}{2}\delta) & \text{for } |x| > \delta \end{cases}$$

Corresponding to a choice of any univariate loss function ($\ell$), like Huber loss as above, we define the DeepONet training loss as follows.

**Definition 4** (A Loss Function for DeepONets). Given $D \subset \mathbb{R}^d$, a compact set with boundary, we define a function class of allowed forcing functions $\mathcal{F} \subset C(D)$. Further, we consider DeepONet maps as given in Definition 1, mapping as $\text{DeepONet} : \mathbb{R}^{d_1} \times \mathbb{R}^{d_2} \to \mathbb{R}$ and consider an instance of the training data given as,

$$\{(f_i, \boldsymbol{x}_{T,i}) \in \mathcal{F} \times \mathbb{R}^{d_2} \mid i = 1, \ldots, \mathcal{S}\}.$$

Then, for an operator $\mathcal{G} : C(D) \to H^s(U)$ where $H^s(U)$ is the $L^2$-based Sobolev space for some $s > 0$, the corresponding DeepONet loss function is given by,

$$\mathcal{L} = \frac{1}{m}\sum_{i=1}^{m} \ell(\mathcal{G}(f_i)(\boldsymbol{x}_{T,i}) - \text{DeepONet}(\boldsymbol{x}_{B,i}, \boldsymbol{x}_{T,i})). \tag{8}$$

where $\ell$ is some loss function. Here we assume a fixed grid of size $d_1$ on which the function $f_i$ gets discretized, to get $\boldsymbol{x}_{B,i} \in \mathbb{R}^{d_1}$.

As the DeepONet function varies, corresponding to it a class of loss functions gets defined via the above equation. This class of DeepONets can be seen as being parameterized by the different choices of weights on the chosen architecture.

Also, it can be seen that the loss function in the experiment described in Section 1.1 was a special case of the loss in equation 8 for $\ell(x) = \frac{x^2}{2}$, the squared loss – and if we assume that the numerical solver exactly solved the forced pendulum O.D.E.

Next, we recall the definition of a key statistical quantity, Rademacher complexity, from Bartlett & Mendelson (2001), which we focus on as our chosen way to measure generalization error for DeepONets.

**Definition 5** (Empirical and Average Rademacher complexity). For a function class $\mathcal{K}$, suppose being given a $m$−sized data-set of points $\{\boldsymbol{x}_i \mid i = 1, \ldots, m\}$ in the domain of the elements in $\mathcal{K}$. For $\epsilon_i \sim \pm 1$ with equal probability, the corresponding empirical Rademacher complexity is given by $\hat{\mathcal{R}}_m(\mathcal{K}) = \mathbb{E}_\epsilon \left[ \sup_{k \in \mathcal{K}} \frac{1}{m} \sum_{i=1}^{m} \epsilon_i k(\boldsymbol{x}_i) \right]$. If the elements of this data-set above are sampled from a distribution $P$, then the average Rademacher complexity is given by,

$$\mathcal{R}_m(\mathcal{K}) = \mathbb{E}_{\boldsymbol{x}_1, \ldots, \boldsymbol{x}_m \sim P} \left( \mathbb{E}_\epsilon \left[ \sup_{k \in \mathcal{K}} \frac{1}{m} \sum_{i=1}^{m} \epsilon_i k(\boldsymbol{x}_i) \right] \right) \tag{9}$$

The crux of our mathematical analysis will be to uncover a recursive structure between the Rademacher complexity of DeepONets and certain DeepONets of one depth lower in the branch and the trunk network, and which would always have one-dimensional outputs for the branch and the trunk and which share weights with the original DeepONet in the corresponding layers.

## 4 Results

Our central result about Rademacher complexity of DeepONets will be stated in Theorem 4.1 of Section 4.1 and consequent to that our result about bounding the generalization error of DeepONets will be stated in Theorem 4.2 of Section 4.2. In Section 4.3, we present an empirical study of our bound on Rademacher complexity for a DeepONet trained to solve the Burgers' P.D.E.

## 4.1 First Main Result : Rademacher Complexity of DeepONets

In the result below we will see how for a certain class of activations, we can get a bound on the Rademacher complexity of DeepONets which does not scale with the width of the nets for the branch and the trunk nets being of arbitrary equal depths.

**Theorem 4.1** (**Rademacher Complexity of Special Symmetric DeepONets**)**.** We consider a special case of DeepONets as given in Definition 1, with (a) $q_B = q_T = n$, and (b) $\sigma_1, \sigma_2$ satisfies Assumption 1 for some constant $L > 0$, and they are positively homogeneous. Further, let $b_{-i}$ and $t_{-i}$ be the number of rows of the weight matrices $\boldsymbol{B}_{n-i}$ and $\boldsymbol{T}_{n-i}$ respectively, as in Definition 2, for $i \geq 1$ and recall from Definition 1 that $p$ is the number of rows of $\boldsymbol{B}_n$ and $\boldsymbol{T}_n$. Then given a class of DeepONet maps as above, we define the following $n-1$ constants, $\mathcal{C}_{n,n-1} > 0$ and $\mathcal{C}_{-k,-k} > 0$, $k = 2, \ldots, n-1$, such that for $\mathcal{S}_k \coloneqq S^{b_{-k}-1} \times S^{t_{-k}-1}$, all the DeepONet maps in the class satisfy the following bounds,

$$\sum_{k_1=1}^{b_{-1}} \sum_{k_2=1}^{t_{-1}} \|\boldsymbol{B}_{n-1,k_1}\| \cdot \|\boldsymbol{T}_{n-1,k_2}\| \cdot \left| \left[ \sum_{j=1}^{p} (\boldsymbol{B}_{n,j} \boldsymbol{T}_{n,j}^{\top}) \right]_{k_1,k_2} \right| \leq \mathcal{C}_{n,n-1},$$

$$\sup_{(\boldsymbol{v},\boldsymbol{w}) \in \mathcal{S}_k} \sum_{j_1=1}^{b_{-k}} \sum_{j_2=1}^{t_{-k}} \left| (\boldsymbol{v}\boldsymbol{w}^{\top})_{j_1,j_2} \right| \|\boldsymbol{B}_{n-k,j_1}\| \|\boldsymbol{T}_{n-k,j_2}\| \leq \mathcal{C}_{-k,-k}, \ \forall k = 2, \ldots, n-1.$$

Then given training data as in Definition 4, the empirical Rademacher complexity of this class is bounded as,

$$\hat{\mathcal{R}}_m \leq \frac{(2L)^{n-1} \mathcal{C}_{n,n-1}}{m} \left( \prod_{j=2}^{n-1} \mathcal{C}_{-j,-j} \right) \sqrt{\sum_{i=1}^{m} \|\boldsymbol{x}_{B,i}\|_2^2 \|\boldsymbol{x}_{T,i}\|_2^2}.$$

Further assuming that the input distribution over $\mathcal{F} \times \mathbb{R}^{d_2}$ induces marginals distributions s.t $\mathbb{E}\left[ \|\boldsymbol{x}_B\|_2^2 \right] \leq M_{x,B}^2, \mathbb{E}\left[ \|\boldsymbol{x}_T\|_2^2 \right] \leq M_{x,T}^2$, we have the average Rademacher complexity of the same class bounded as,

$$\mathcal{R}_m \leq \frac{(2L)^{n-1} \mathcal{C}_{n,n-1}}{\sqrt{m}} \left( \prod_{j=2}^{n-1} \mathcal{C}_{-j,-j} \right) M_{x,B} M_{x,T}.$$

The proof for the above is outlined in Section 6. For $\sigma_1(x) = \sigma_2(x) = |x|$ i.e. for DeepONets with absolute value activations, we have $L = 1$ and hence the subsequent simplification also happens in the result in Theorem 4.1.

Now we have all the necessary setup to state the final result on generalization bound for DeepONets.

## 4.2 Second Main Result : Size-Independent Generalization Error Bound for DeepONets Trained via a Huber Loss on Unbounded Data

**Theorem 4.2** (Generalization Error Bound for DeepONet)**.** We continue with the setup of the DeepONets as in Theorem 4.1 but with $\sigma_1(x) = \sigma_2(x) = |x|$. For an operator $\mathcal{G} : C(D) \to H^s(U)$ where $H^s(U)$ is the $L^2$-based Sobolev space for some $s > 0$, a class of forcing functions $\mathcal{F} \subset C(D)$ and $\mathcal{W}$ a set of possible weights $\boldsymbol{w}$ for DeepONets (denoted as DeepONet$_{\boldsymbol{w}}$), we define the function class $\mathcal{H}$ as,

$$\mathcal{H} \coloneqq \{(f, \mathbf{x}_T) \mapsto \mathcal{G}(f)(\mathbf{x}_T) - \text{DeepONet}_{\boldsymbol{w}}(\mathbf{x}_B(f), \mathbf{x}_T) \,|\, \boldsymbol{w} \in \mathcal{W}, (f, \mathbf{x}_T) \in \mathcal{F} \times \mathbb{R}^{d_2},$$
$$\mathcal{G}(f)(\mathbf{x}_T), \text{DeepONet}_{\boldsymbol{w}}(\mathbf{x}_B(f), \mathbf{x}_T) \in \mathbb{R}\}$$

Then for Huber loss $\ell_{H,\delta}(\cdot)$ as in Definition 3 with $\delta = \left(\frac{1}{2}\right)^{n-1}$ where $n$ is the depth of the branch and the trunk nets we have the following generalization bound,

$$\mathbb{E}_{\{(f_i, \boldsymbol{x}_{T,i})^{iid}_{\sim}\mathcal{D} \,|i=1\ldots m\}} \left[ \sup_{\mathbf{w} \in \mathcal{W}} \left[ \frac{1}{m} \sum_{i=1}^{m} \ell_{H,\delta}(h(f_i, \boldsymbol{x}_{T,i})) - \mathbb{E}_{(f, \boldsymbol{x}_T) \sim \mathcal{D}} \left[ \ell_{H,\delta}(h(f, \boldsymbol{x}_T)) \right] \right] \right] \tag{10}$$
$$\leq \frac{2\mathcal{C}_{n,n-1}}{\sqrt{m}} \left( \prod_{j=2}^{n-1} \mathcal{C}_{-j,-j} \right) M_{x,B} M_{x,T}$$

where $\mathcal{C}_{n,n-1}$ and $\mathcal{C}_{-j,-j}$ are the constants as defined in Theorem 4.1, and $\mathcal{D}$ is a distribution over $\mathcal{F} \times \mathbb{R}_{d_2}$.

The proof for the above theorem is given in Section 6.

### 4.3 An Experimental Exploration of the Proven Rademacher Complexity Bound for DeepONets

In the following sections, we present an experimental study on the complexity bounds for the Burgers' and 2D-Heat PDEs in Sections 4.3.1 and 4.3.2, respectively.

#### 4.3.1 Burgers' PDE

Here, we consider the following specification of the Burgers' P.D.E. with periodic boundary conditions,

$$\frac{\partial s}{\partial t} + s\frac{\partial s}{\partial x} = \kappa\frac{\partial^2 s}{\partial x^2}, \quad (x,t) \in [-\pi, \pi] \times [0, \mathrm{T}]$$

$$s(x - \pi, t) = s(x + \pi, t), \ s(x, 0) = u(x)$$

where, $\kappa > 0$ denotes the fluid viscosity, for our experiments we choose $\kappa = 0.01$, and $u(x)$ is a $2\pi$-periodic initial condition with zero mean i.e. $\int_{-\pi}^{\pi} u(x)dx = 0$ and $\mathrm{T} \in \mathbb{R}$.

Hence it follows that the solution operator of Definition 4 corresponding to the above maps the initial condition $u$ to the solution to the Burgers' P.D.E., $s$. Hence, we will approximate the implicit solution operator $\mathcal{G}$ with a DeepONet $\tilde{\mathcal{G}}$ — which for this case would be a map as follows,

$$\mathbb{R}^m \times \mathbb{R}^2 \to \mathbb{R}, \ (\boldsymbol{v}, \boldsymbol{y}) \mapsto \tilde{\mathcal{G}}(\boldsymbol{v}, \boldsymbol{y}) := \sum_{k=1}^{q} \mathcal{N}_{\mathrm{B,k}}(\boldsymbol{v}) \cdot \mathcal{N}_{\mathrm{T,k}}(\boldsymbol{y})$$

where $\mathcal{N}_{\mathrm{B}}$ and $\mathcal{N}_{\mathrm{T}}$ are any two neural nets mapping $\mathbb{R}^m \to \mathbb{R}^q$ and $\mathbb{R}^2 \to \mathbb{R}^q$ respectively - the Branch net and the Trunk net. In above, $q$, the common output dimension of the branch and the trunk net is a hyperparameter for the experiment.

The data required to set up the training for the above map $\tilde{G}$ can be summarized as a 3-tuple of tensors, $(u_{\mathrm{training}}, y_{\mathrm{training}}, s_{\mathrm{training}})$ of dimensions and descriptions as given below.

- $u_{\mathrm{training}} \in \mathbb{R}^{N_{\mathrm{training}} \times m}$ are evaluations of the $N_{\mathrm{training}}$ number of input functions at the $m$ sensors. We denote the $i^{th}$ row of the above as, $[u_i(x_1), u_i(x_2), \ldots, u_i(x_m)] \ \forall i \in \{1, \ldots, N_{\mathrm{training}}\}$

- $y_{\mathrm{training}} \in \mathbb{R}^{N_{\mathrm{training}} \times P_{\mathrm{training}} \times 2}$ are the $P_{\mathrm{training}}$ uniformly sampled collocation points in $\mathbb{R}^2$ corresponding to each of the $N_{\mathrm{training}}$ number of input functions. And $\forall i \in \{1, \ldots, N_{\mathrm{training}}\}$, we denote the sampled points as $\{(x_{\mathrm{i,j}}, t_{\mathrm{i,j}}) \mid j = 1, \ldots, P_{\mathrm{training}}\} \overset{\mathrm{uniform}}{\sim} ([-\pi, \pi] \times [0, T])^{P_{\mathrm{training}}}$

- $s_{\mathrm{training}} \in \mathbb{R}^{N_{\mathrm{training}} \times P_{\mathrm{training}}}$ are the values of the numerical solution of the P.D.E. at the respective collocation points. And the $(i, j)^{\mathrm{th}}$-entry of this would be denoted as $s_{\mathrm{i}}(x_{\mathrm{i,j}}, t_{\mathrm{i,j}})$.

In terms of the above training data, the corresponding training loss (Definition 4) is,

$$\mathcal{L}_{\mathrm{Training}}(\theta) = \frac{1}{N_{\mathrm{training}}P_{\mathrm{training}}} \sum_{i=1}^{N_{\mathrm{training}}} \sum_{j=1}^{P_{\mathrm{training}}} \ell\Big(s_i(x_{\mathrm{i,j}}, t_{\mathrm{i,j}})$$
$$-\sum_{k=1}^{q} \mathcal{N}_{\mathrm{B,k}}(u_i(x_1), u_i(x_2), \ldots, u_i(x_m)) \cdot \mathcal{N}_{\mathrm{T,k}}(x_{\mathrm{i,j}}, t_{\mathrm{i,j}})\Big) \tag{11}$$

whereby in our experiments $\ell$ is either chosen as the $\ell_2$ loss or the Huber loss and $\theta$ denotes the vector of all trainable parameters over both the nets. Similarly as above the test loss $\mathcal{L}_{\mathrm{Test}}$ would be defined corresponding to a random sampling of $N_{\mathrm{test}}$ input functions and $P_{\mathrm{test}}$ points in the domain of the P.D.E. where the true solution is known corresponding to each of these test input functions. For our experiments, we fixed $N_{training}$ to 400 and $P_{training}$ is chosen to be the following set of values $\{300, 400, 500, 750, 1000, 1500, 2000\}$.

Further, for our experiments, we sampled the input functions $u_i$ by sampling functions of the form, $u_i(x) = \sum_{n=1}^{N} c_n \sin((n+1)x)$, where the coefficients were sampled as, $c_n \sim \mathcal{N}(0, A^2)$, with $A$ and $N$ being

arbitrarily chosen constants. This way of sampling ensures that the functions chosen are all $2\pi$-periodic and will have zero mean in the interval $[-\pi, \pi]$ - as required in our setup.

In this section, we demonstrate our Rademacher complexity bound by training DeepONets on the above loss function and measuring the generalization gap of the trained DeepONet obtained and computing for this trained DeepONet an upperbound on the essential part of our Rademacher bound (i.e. $\frac{\mathcal{C}_{n,n-1}}{\sqrt{m}}\left(\prod_{j=2}^{n-1}\tilde{\mathcal{C}}_{-j,-j}\right)$ where $\tilde{\mathcal{C}}_{-k,k}$ is as defined in equation 5). Then we vary the number of training data (i.e. $N_{\text{training}} \times P_{\text{training}}$) and show that the two numbers computed above are significantly correlated as this training hyperparameter is varied. We use branch and trunk nets each of which are of depth 3 and width 100.

The correlation plots shown in Figures 2a and 2b, correspond to training on the Huber loss function, with two different values of $\delta$ - and the former corresponds to the special value of $\delta = \frac{1}{4}$ where the size-independent generalization bound in Theorem 4.2 clicks for this setup.

In Figure 6 of Appendix, we repeat the same experiments but for the $\ell_2$ loss function and show that the required correlation persists even with this loss which is outside the scope of Theorem 4.2.

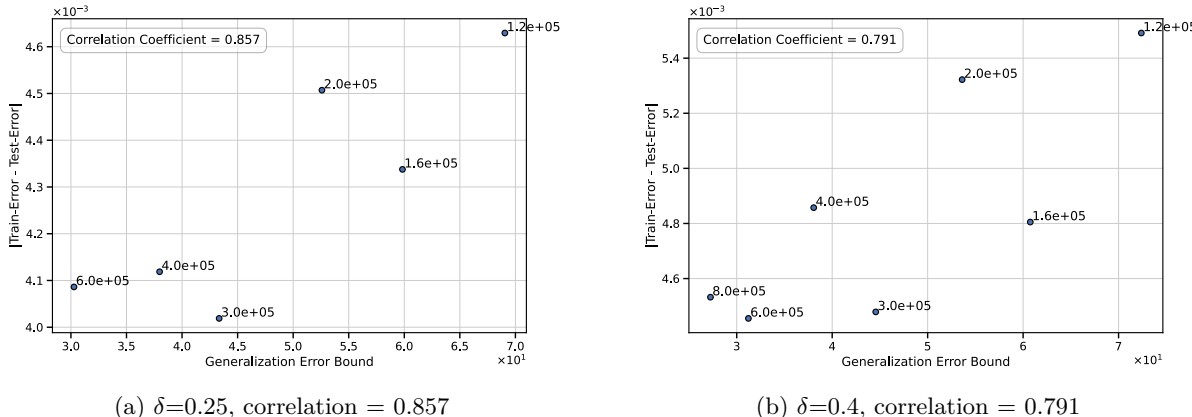

(a) $\delta$=0.25, correlation = 0.857

(b) $\delta$=0.4, correlation = 0.791

Figure 2: The above plot shows the behaviour of the measured generalization error with respect to $\frac{\mathcal{C}_{3,2}\tilde{\mathcal{C}}_{-2,-2}}{\sqrt{m}}$ for training DeepONets, to solve the Burgers' PDE, with empirical loss as given in equation 11, specialized to the Huber loss (Definition 3) for the stated values of $\delta$ and for the branch and the trunk nets being of depth 3. Each point is labelled by the number of training data used in that experiment.

### 4.3.2 Heat PDE

Here, we consider the following specification of the 2D-Heat P.D.E. (Altaisan, 2024),

$$\frac{\partial u}{\partial t} = \kappa\nabla^2 u = \kappa\left(\frac{\partial^2 u}{\partial x^2} + \frac{\partial^2 u}{\partial y^2}\right), \quad (x,y) \in D, \quad t \in [0,T]$$
$$u(x,y,t) = 0, \quad (x,y) \in \partial D, \quad u(x,y,0) = f(x,y), \quad (x,y) \in D$$

where, $D$ is the 2-dimensional square $[0,1]\times[0,1]$, $\kappa > 0$ denotes the thermal diffusivity, for our experiments we choose $\kappa = 1$, and $f(x,y)$ is a $2\pi$-periodic initial condition and $T \in \mathbb{R}$.

In this case, the solution operator of Definition 4 corresponding to the above maps the initial condition $f$ to the solution to the Heat P.D.E, $s$. Hence, we will approximate the implicit solution operator $\mathcal{G}$ with a DeepONet $\tilde{\mathcal{G}}$ — which for this case would be a map as follows,

$$\mathbb{R}^m \times \mathbb{R}^3 \to \mathbb{R}, \ (\boldsymbol{v}, \boldsymbol{y}) \mapsto \tilde{\mathcal{G}}(\boldsymbol{v}, \boldsymbol{y}) \coloneqq \sum_{k=1}^{q} \mathcal{N}_{\text{B},k}(\boldsymbol{v}) \cdot \mathcal{N}_{\text{T},k}(\boldsymbol{y})$$

where $\mathcal{N}_{\text{B}}$ and $\mathcal{N}_{\text{T}}$ are any two neural nets mapping $\mathbb{R}^m \to \mathbb{R}^q$ and $\mathbb{R}^3 \to \mathbb{R}^q$ respectively - the Branch net and the Trunk net. In above, $q$, the common output dimension of the branch and the trunk net is a hyperparameter for the experiment. We have chosen $q = 128$ and $m = 100$ for our experiments.

The data required to set up the training for the above map $\tilde{G}$ can be summarized as a 3–tuple of tensors, $(f_{\text{training}},\ p_{\text{training}},\ u_{\text{training}})$ of dimensions and descriptions as given below.

- $f_{\text{training}} \in \mathbb{R}^{N_{\text{training}} \times m}$ are evaluations of the $N_{\text{training}}$ number of input functions at the $m$ sensors. We denote the $i^{th}$ row of the above as, $[f_i(x_1, y_1), f_i(x_2, y_2), \ldots, f_i(x_{\text{m}}, y_{\text{m}})]\ \forall i \in \{1, \ldots, N_{\text{training}}\}$

- $p_{\text{training}} \in \mathbb{R}^{N_{\text{training}} \times P_{\text{training}} \times 3}$ are $P_{\text{training}}$ uniformly sampled collocation points in $\mathbb{R}^3$ corresponding to each of the $N_{\text{training}}$ number of input functions. And $\forall i \in \{1, \ldots, N_{\text{training}}\}$, we denote the sampled points as $\{(x_{\text{i},\text{j}}, y_{\text{i},\text{j}}, t_{\text{i},\text{j}}) \mid j = 1, \ldots, P_{\text{training}}\} \overset{\text{uniform}}{\sim} ([0,1] \times [0,1] \times [0,T])^{P_{\text{training}}}$

- $u_{\text{training}} \in \mathbb{R}^{N_{\text{training}} \times\ P_{\text{training}}}$ are the values of the numerical solution of the P.D.E. at the respective collocation points. And the $(i, j)^{\text{th}}$–entry of this would be denoted as $u_{\text{i}}(x_{\text{i},\text{j}}, y_{\text{i},\text{j}}, t_{\text{i},\text{j}})$.

In terms of the above training data, the corresponding training loss (Definition 4) is,

$$\mathcal{L}_{\text{Training}}(\theta) = \frac{1}{N_{\text{training}}P_{\text{training}}} \sum_{i=1}^{N_{\text{training}}} \sum_{j=1}^{P_{\text{training}}} \ell\left(u_i(x_{\text{i},\text{j}}, y_{\text{i},\text{j}}, t_{\text{i},\text{j}})\right.$$
$$\left. - \sum_{k=1}^{q} \mathcal{N}_{\text{B,k}}(f_i(x_1, y_1), f_i(x_2, y_2), \ldots, f_i(x_m, y_m)) \cdot \mathcal{N}_{\text{T,k}}(x_{\text{i},\text{j}}, y_{\text{i},\text{j}}, t_{\text{i},\text{j}})\right) \quad (12)$$

whereby in our experiments $\ell$ is either chosen as the $\ell_2$ loss or the Huber loss and $\theta$ denotes the vector of all trainable parameters over both the nets. Similarly as above the test loss $\mathcal{L}_{\text{Test}}$ would be defined corresponding to a random sampling of $N_{\text{test}}$ input functions and $P_{\text{test}}$ points in the domain of the P.D.E. where the true solution is known corresponding to each of these test input functions. For our experiments, we fixed $N_{training}$ to 64 and $P_{training}$ is chosen to be the following set of values $\{2^6, 3^6, \ldots, 7^6\}$.

Further, for our experiments, we sampled the input functions $f_i$ by sampling functions of the form, $f_i(x, y) = \left[\sum_{m=1}^{N} c_m \sin(\frac{m\pi x}{x_0})\right] \cdot \left[\sum_{n=1}^{N} d_n \sin(\frac{n\pi y}{y_0})\right]$, where the coefficients were sampled as, $c_m, d_n \sim U([100, 200])$, with $N$ is fixed to 2. This way of sampling ensures that the functions chosen are all $2\pi$-periodic and we know the exact analytical solution for these initial conditions to be $u(x, y, t) = \sum_{m=1}^{N} \sum_{n-1}^{N} A_{mn} \sin\left(\frac{m\pi x}{x_0}\right) \sin\left(\frac{n\pi y}{y_0}\right) e^{-\pi^2\left(\frac{m^2}{x_0^2} + \frac{n^2}{y_0^2}\right)t}$, where $N = 2$, $A_{mn} = c_m\ d_n$ and $x_0, y_0 = 1$. This exactness leads to a more controlled measure of the test error than in the previous setup.

Here, we demonstrate our Rademacher complexity bound by training DeepONets on the above loss function and measuring the generalization gap of the trained DeepONet obtained and computing for this trained DeepONet an upper-bound on the essential part of our Rademacher bound (i.e. $\frac{\mathcal{C}_{n,n-1}}{\sqrt{m}}\left(\prod_{j=2}^{n-1} \tilde{\mathcal{C}}_{-j,-j}\right)$ where $\tilde{\mathcal{C}}_{-k,k}$ is as defined in equation 5). Then we vary the number of training data (i.e. $N_{\text{training}} \times P_{\text{training}}$) and show that the two numbers computed above are significantly correlated as this training hyperparameter is varied. We use branch and trunk nets each of which are of depth 3 and width 128.

The correlation plots shown in Figures 3a and 3b, correspond to training on the Huber loss function, with two different values of $\delta$ - and the former corresponds to the special value of $\delta = \frac{1}{4}$ where the size-independent generalization bound in Theorem 4.2 clicks for this setup.

Going beyond the ambit of the Rademacher bound proven, in Figure 7 in Appendix M, we present the plots demonstrating the solution of the Heat P.D.E. during training with $\ell_2$ and Huber loss, while allowing for ReLU activation and biases in the layers. It can be seen that Huber loss demonstrates noticeably better performance compared to $\ell_2$ loss.

## 5 Discussion

The most immediate research direction that is being suggested by our generalization error bound is to explore if the advantages shown here for Huber loss can make DeepONets competitive with CNO (Raonić et al., 2023).

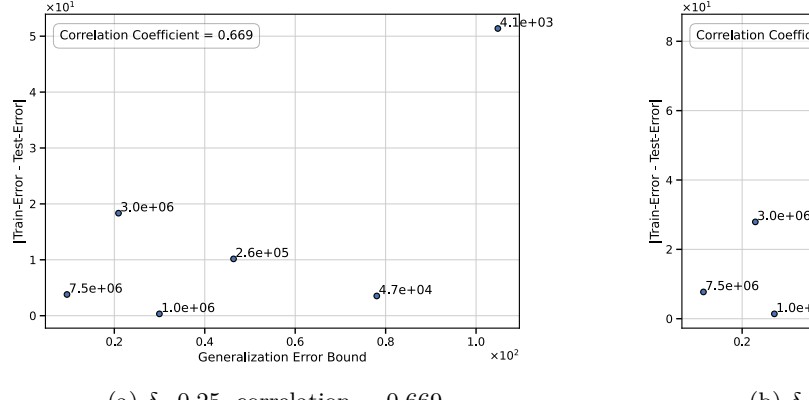 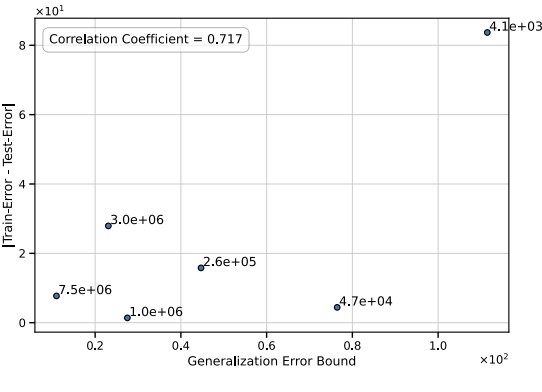

(a) $\delta$=0.25, correlation = 0.669          (b) $\delta$=0.4, correlation = 0.717

Figure 3: The above plot shows the behaviour of the measured generalization error with respect to $\frac{\mathcal{C}_{3,2}\tilde{\mathcal{C}}_{-2,-2}}{\sqrt{m}}$ for training DeepONets, to solve the 2D-Heat PDE, with empirical loss as given in equation 12, specialized to the Huber loss (Definition 3) for the stated values of $\delta$ and for the branch and the trunk nets being of depth 3. Each point is labelled by the number of training data used in that experiment.

Further, in Wang et al. (2021), an unsupervised variation of the loss function of a DeepONet was shown to give better performance. In Goswami et al. (2022b), authors employ a variational framework for solving differential equations using DeepONets, through a novel loss function. Understanding precisely when these variations in the loss function used give advantages over the basic DeepONet loss is yet another interesting direction for future research - and the path towards such goals could be to explore if the methods demonstrated here for computation of generalization bounds can be used for these novel setups too.

Also, it would be fruitful to understand how the generalization bounds obtained in this work can be modified to cater to the variations of the DeepONet architecture (Kontolati et al., 2023), (Bonev et al., 2023) Lu et al. (2022) that are getting deployed.

In the context of understanding the generalization error of Siamese networks, the authors in Dahiya et al. (2021) dealt with a certain product of neural outputs structure (where the nets share weights). In their analysis, the authors bound the Rademacher complexity via covering numbers. Since we try to directly bound the Rademacher complexity for DeepONets, it would be interesting to investigate if our methods can be adapted to improve such results about Siamese nets.

Lastly, we note that the existing bounds on the Rademacher complexity of nets have typically been proven by making ingenious use of the algebraic identities satisfied by Rademacher complexity (Theorem 12 in Bartlett & Mendelson (2001)). But to the best of our knowledge, we are not aware of any general result on how Rademacher complexity of a product of function spaces can be written in terms of individual Rademacher complexities. We posit that such a mathematical development, if achieved in generality, can be a significant advance affecting various fields.

## 6  Methods

In this section, we will outline the proofs of the main results, Theorems 4.1 and 4.2. We would like to emphasize that the subsequent propositions 6.2 and 6.3 hold in more generality than Theorem 4.1, because they do not need the branch and the trunk nets to be of equal depth.

**Outline of the Proof Techniques**  Derivation of the first main result, Theorem 4.1, involves 3 key steps: **(a)** formulating a variation of the standard Talagrand contraction (Lemma 6.1) **(b)** using this to bound the Rademacher complexity of a class of DeepONets with certain activations (e.g. absolute value) by the Rademacher complexity for a class of DeepONets having one less depth and 1−dimensional outputs, for both the branch and the trunk (Lemma 6.2) and lastly **(c)** uncovering a recursive structure for the

Rademacher complexity across depths between special DeepONet classes having 1–dimensional outputs for both the branch and the trunk. (Lemma 6.3).

Lemma 6.2 removes the last 4 matrices from the DeepONet (2 each from the branch and the trunk) leading to one-dimensional output branch and trunk nets. Lemma 6.3 removes 2 matrices (1 each from branch and trunk) – by an entirely different argument than needed in the former. Lemma 6.2 is invoked only once at the beginning, while Lemma 6.3 is repeatedly used for each remaining layer of the DeepONet.

We note that both our "peeling" lemmas above are structurally very different from the one in Golowich et al. (2018) - where the last layer of a standard net gets peeled in every step.

Deriving the second main result, Theorem 4.2, involves the following key steps: **(a)** establishing a relationship between the Rademacher complexity of the loss class and that of the DeepONet when the loss function is Lipschitz (Proposition G.6) and **(b)** using this to upper bound the expectation over data of the supremum of generalization error in terms of the Rademacher complexity of the DeepONet.

Towards proving Theorem 4.1, we need the following lemma which can be seen as a variation of the standard Talagrand contraction lemma,

**Lemma 6.1.** Let $\phi_P, \phi_Q : \mathbb{R} \to \mathbb{R}$ be two functions such that Assumption 1 holds and let $\mathcal{P}$ and $\mathcal{Q}$ be any two sets of real-valued functions. Then given any two sets of points $\{\boldsymbol{x}_i \mid i = 1, \ldots, m\}$ and $\{\boldsymbol{y}_i \mid i = 1, \ldots, m\}$ in the domains of the functions in $\mathcal{P}$ and $\mathcal{Q}$ respectively, we have the following inequality of Rademacher complexities - where both the sides are being evaluated on this same set of points,

$$\mathcal{R}_{\mathrm{m}}(\phi_P \circ \mathcal{P} \cdot \phi_Q \circ \mathcal{Q}) \le L\mathcal{R}_{\mathrm{m}}(\mathcal{P} \cdot \mathcal{Q}).$$

The above lemma has been proven in Appendix H.1.

Towards stating Propositions 6.2 and 6.3, we will need to define certain classes of sub-DeepONets s.t these sub-DeepONets would be one depth lower in the branch and trunk network, and would always have one dimensional outputs for both the branch and trunk nets and would share weights below that with the corresponding layers in the original DeepONet.

**Definition 6. (Classes of sub-DeepONets)** Let $\mathcal{W}_{\mathrm{rest}}$ be a set of allowed matrices for nets $f'_B$ and $f'_T$ as in Definition 2. Now, given a constant $\mathcal{C}_{q_B, q_T, q_B-1, q_T-1} > 0$ we define the following set of 4–tuples of outermost layer of matrices in the DeepONet as,

$$\mathcal{W}(\mathcal{C}_{q_B, q_T, q_B-1, q_T-1}) := \left\{ (\boldsymbol{B}_{q_B}, \boldsymbol{B}_{q_B-1}, \boldsymbol{T}_{q_T}, \boldsymbol{T}_{q_T-1}) \; \middle| \right.$$
$$\left. \sum_{k_1=1}^{b_{-1}} \sum_{k_2=1}^{t_{-1}} \|\boldsymbol{B}_{q_B-1, k_1}\| \cdot \|\boldsymbol{T}_{q_T-1, k_2}\| \cdot \left| \left[ \sum_{j=1}^{p} (\boldsymbol{B}_{q_B, j} \boldsymbol{T}_{q_T, j}^{\top}) \right]_{k_1, k_2} \right| \le \mathcal{C}_{q_B, q_T, q_B-1, q_T-1} \right\}. \quad (13)$$

Secondly, corresponding to the above we define the following class of DeepONets,

$$\mathrm{DeepONet}(\mathcal{W}(\mathcal{C}_{q_B, q_T, q_B-1, q_T-1}), \mathcal{W}_{\mathrm{rest}}) :=$$
$$\left\{ \mathrm{DeepONet}_{\boldsymbol{w}} \text{ (as in Definition 1)} \; \middle| \; \boldsymbol{w} \in (\mathcal{W}(\mathcal{C}_{q_B, q_T, q_B-1, q_T-1}), \mathcal{W}_{\mathrm{rest}}) \right\}. \quad (14)$$

Lastly, we define the following class of DeepONets,

$$\mathrm{DeepONet}(\mathcal{W}_{\mathrm{rest}}) := \left\{ (\boldsymbol{x}_B, \boldsymbol{x}_T) \mapsto \boldsymbol{v}^{\top} \boldsymbol{f}'_B(\boldsymbol{x}_B) \cdot \boldsymbol{w}^{\top} \boldsymbol{f}'_T(\boldsymbol{x}_T) \in \mathbb{R} \right.$$
$$\left. \middle| \text{ the set of allowed matrices for nets } \boldsymbol{f}'_B \text{ and } \boldsymbol{f}'_T \text{ are in } \mathcal{W}_{\mathrm{rest}} \; ; \; (\boldsymbol{v}, \boldsymbol{w}) \in S^{b-2-1} \times S^{t-2-1} \right\}. \quad (15)$$

**Proposition 6.2** (**Removal of the Last** $4$ **Matrices of a DeepONet**)**.** We continue to be in the setup of Definition 2 and assuming that $\sigma_1, \sigma_2$ satisfies Assumption 1 for some $L > 0$, and are positively homogeneous. Then given the definitions of DeepONet$(\mathcal{W}(\mathcal{C}_{q_B, q_T, q_B-1, q_T-1}), \mathcal{W}_{\text{rest}})$ and DeepONet$(\mathcal{W}_{\text{rest}})$ in Definition 6, we have the following upperbound on empirical Rademacher complexity of a DeepONet, (here $\mathcal{S}_k := S^{b_{-k}-1} \times S^{t_{-k}-1}$)

$$\hat{\mathcal{R}}_m\Big(\text{DeepONet}(\mathcal{W}(\mathcal{C}_{q_B, q_T, q_B-1, q_T-1}), \mathcal{W}_{\text{rest}})\Big)$$

$$\leq 2L \cdot \mathcal{C}_{q_B, q_T, q_B-1, q_T-1} \cdot \mathbb{E}_\epsilon\Bigg[\sup_{\substack{(\boldsymbol{v}, \boldsymbol{w}) \in \mathcal{S}_2 \\ \mathcal{W}_{\text{rest}}}}\Bigg(\sum_{i=1}^m \epsilon_i \boldsymbol{v}^\top \boldsymbol{f}'_B(\boldsymbol{x}_{B,i}) \cdot \boldsymbol{w}^\top \boldsymbol{f}'_T(\boldsymbol{x}_{T,i})\Bigg)\Bigg]$$

$$\leq 2L \cdot \mathcal{C}_{q_B, q_T, q_B-1, q_T-1} \cdot \hat{\mathcal{R}}_m\Big(\text{DeepONet}(\mathcal{W}_{\text{rest}})\Big).$$

Note that both sides of the above are computed for the same data $\{(\boldsymbol{x}_{B,i}, \boldsymbol{x}_{T,i}) \in \mathbb{R}^{d_1} \times \mathbb{R}^{d_2} \mid i = 1, \ldots, m\}$. The proof of the above proposition is given in Appendix E.

Referring to the definitions of the DeepONet classes on the L.H.S. and the R.H.S. of the above, as given in equations 14 and 15 respectively, we see that the above lemma upperbounds the Rademacher complexity of a DeepONet class (whose individual nets can have multi-dimensional outputs) by the Rademacher complexity of a simpler DeepONet class. The DeepONet class in the R.H.S. is simpler because the last layer of each of the individual nets therein is constrained to be a unit vector of appropriate dimensions (and thus the individual nets here are always of 1 dimensional output) – and whose both branch and the trunk are shorter in depth by 1 activation and 1 linear transform than in the L.H.S.

**Proposition 6.3** (**Peeling for DeepONets**)**.** We continue in the setup of Proposition 6.2 and define the functions $\boldsymbol{f}''_B$ and $\boldsymbol{f}''_T$ s.t we have the following equalities, $\boldsymbol{f}'_B = \sigma_1\left(\boldsymbol{B}_{q_B-2}\boldsymbol{f}''_B\right)$ and $\boldsymbol{f}'_T = \sigma_2\left(\boldsymbol{T}_{q_T-2}\boldsymbol{f}''_T\right)$ Further, given a constant $\mathcal{C}_{-2,-2} > 0$, we define $\mathcal{W}'_{rest}$ as the union of (a) the set of weights that are allowed in the $\mathcal{W}_{rest}$ set for the matrices $\boldsymbol{B}_{q_B-3}, \boldsymbol{B}_{q_B-4}, \ldots, \boldsymbol{B}_1$ and $\boldsymbol{T}_{q_T-3}, \boldsymbol{T}_{q_T-4}, \ldots, \boldsymbol{T}_1$ and (b) the subset of the weights for $\boldsymbol{B}_{q_B-2}$ and $\boldsymbol{T}_{q_T-2}$ that are allowed by $\mathcal{W}_{rest}$ which also additionally satisfy the constraint, (here $\mathcal{S}_k := S^{b_{-k}-1} \times S^{t_{-k}-1}$)

$$\sup_{(\boldsymbol{v}, \boldsymbol{w}) \in \mathcal{S}_2} \sum_{j_1=1}^{b_{-2}} \sum_{j_2=1}^{t_{-2}} \big|(\boldsymbol{v}\boldsymbol{w}^\top)_{j_1, j_2}\big| \|\boldsymbol{B}_{q_B-2, j_1}\| \|\boldsymbol{T}_{q_T-2, j_2}\| \leq \mathcal{C}_{-2, -2}. \tag{16}$$

Then we get the following inequality between Rademacher complexities,

$$\mathbb{E}_\epsilon\Bigg[\sup_{\substack{(\boldsymbol{v}, \boldsymbol{w}) \in \mathcal{S}_2 \\ \mathcal{W}_{\text{rest}}}}\Bigg(\sum_{i=1}^m \epsilon_i \boldsymbol{v}^\top \boldsymbol{f}'_B(\boldsymbol{x}_{B,i}) \cdot \boldsymbol{w}^\top \boldsymbol{f}'_T(\boldsymbol{x}_{T,i})\Bigg)\Bigg]$$

$$\leq 2L\mathcal{C}_{-2,-2}\mathbb{E}_\epsilon\Bigg[\sup_{\substack{(\boldsymbol{v}, \boldsymbol{w}) \in \mathcal{S}_3 \\ \mathcal{W}'_{rest}}}\Bigg(\sum_{i=1}^m \epsilon_i(\boldsymbol{v}^\top \boldsymbol{f}''_B(\boldsymbol{x}_{B,i}))(\boldsymbol{w}^\top \boldsymbol{f}''_T(\boldsymbol{x}_{T,i}))\Bigg)\Bigg],$$

where $b_{-k}, t_{-k}$ are as in Definition 2.

Proof of the above proposition is given in Appendix F. And now we have stated all the intermediate results needed to prove our bounds on the Rademacher complexity of a certain DeepONet class.

**Proof of Theorem 4.1**

*Proof.* For each $k = 2, 3, \ldots, n-1$, we define a product of unit-spheres, $\mathcal{S}_k := S^{b_{-k}-1} \times S^{t_{-k}-1}$ and let $\mathcal{S}_n := S^{d_1-1} \times S^{d_2-1}$. Now we define,

$$\mathcal{W}^1_{\text{rest}} := \Bigg\{(\boldsymbol{B}_n, \boldsymbol{B}_{n-1}, \boldsymbol{T}_n, \boldsymbol{T}_{n-1}) \ \bigg| \ \sum_{k_1=1}^{b_{-1}} \sum_{k_2=1}^{t_{-1}} \|\boldsymbol{B}_{n-1, k_1}\| \cdot \|\boldsymbol{T}_{n-1, k_2}\| \cdot \Bigg[\sum_{j=1}^p (\boldsymbol{B}_{n,j} \boldsymbol{T}_{n,j}^\top)\Bigg]_{k_1, k_2} \leq \mathcal{C}_{n, n-1}\Bigg\}.$$

Next, for each $i = 2, 3, \ldots, n-1$, we define

$$\mathcal{W}_{\text{rest}}^i := \bigcup_{k=1}^{n-i} \left\{ (\boldsymbol{B}_k, \boldsymbol{T}_k) \;\middle|\; \sup_{(\boldsymbol{v}, \boldsymbol{w}) \in \mathcal{S}_k} \sum_{j_1=1}^{b_{-k}} \sum_{j_2=1}^{t_{-k}} \left| (\boldsymbol{v}\boldsymbol{w}^\top)_{j_1, j_2} \right| \cdot \| \boldsymbol{B}_{n-k, j_1} \| \cdot \| \boldsymbol{T}_{n-k, j_2} \| \le \mathcal{C}_{-k, -k} \right\}.$$

Thus we have,

$$m\hat{\mathcal{R}}_m = \mathbb{E}_\epsilon \left[ \sup_{(\mathcal{W}_{\text{rest}}^1, \mathcal{W}_{\text{rest}}^2)} \left( \sum_{i=1}^m \epsilon_i \left\langle \boldsymbol{B}_n \sigma_1 \left( \boldsymbol{B}_{n-1} \boldsymbol{f}_B'(\boldsymbol{x}_{B,i}) \right), \boldsymbol{T}_n \sigma_2 \left( \boldsymbol{T}_{n-1} \boldsymbol{f}_T'(\boldsymbol{x}_{T,i}) \right) \right\rangle \right) \right].$$

Then we can invoke Lemma 6.2 on the above to get,

$$m\hat{\mathcal{R}}_m = \mathbb{E}_\epsilon \left[ \sup_{(\mathcal{W}_{\text{rest}}^1, \mathcal{W}_{\text{rest}}^2)} \left( \sum_{i=1}^m \epsilon_i \left\langle \boldsymbol{B}_n \sigma_1 \left( \boldsymbol{B}_{n-1} \boldsymbol{f}_B'(\boldsymbol{x}_{B,i}) \right), \boldsymbol{T}_n \sigma_2 \left( \boldsymbol{T}_{n-1} \boldsymbol{f}_T'(\boldsymbol{x}_{T,i}) \right) \right\rangle \right) \right]$$

$$\le 2L\mathcal{C}_{n,n-1} \mathbb{E}_\epsilon \left[ \sup_{\substack{\mathcal{W}_{\text{rest}}^2 \\ \boldsymbol{v}_1, \boldsymbol{v}_2 \in \mathcal{S}_2}} \left( \sum_{i=1}^m \epsilon_i \boldsymbol{v}_1^\top \boldsymbol{f}_B'(\boldsymbol{x}_{B,i}) \, \boldsymbol{v}_2^\top \boldsymbol{f}_T'(\boldsymbol{x}_{T,i}) \right) \right].$$

Now, using Lemma 6.3 repeatedly on the R.H.S above, and defining in a natural fashion the subsequent branch and trunk sub-networks as $\boldsymbol{f}^{(i)}(\cdot)$ we have,

$$m\hat{\mathcal{R}}_m \le (2L)^2 \mathcal{C}_{n,n-1} \mathcal{C}_{-2,-2} \mathbb{E}_\epsilon \left( \sup_{\substack{\mathcal{W}_{rest}^3 \\ (\boldsymbol{v}_1, \boldsymbol{v}_2) \in \mathcal{S}_3}} \sum_{i=1}^m \epsilon_i (\boldsymbol{v}_1^\top \boldsymbol{f}_B''(\boldsymbol{x}_{B,i}))(\boldsymbol{v}_2^\top \boldsymbol{f}_T''(\boldsymbol{x}_{T,i})) \right)$$

$$\vdots$$

$$\le (2L)^i \mathcal{C}_{n,n-1} \left( \prod_{j=2}^i \mathcal{C}_{-j,-j} \right) \mathbb{E}_\epsilon \left( \sup_{\substack{\mathcal{W}_{\text{rest}}^{i+1} \\ (\boldsymbol{v}_1, \boldsymbol{v}_2) \in \mathcal{S}_{i+1}}} \sum_{i=1}^m \epsilon_i (\boldsymbol{v}_1^\top \boldsymbol{f}_B^{(i)}(\boldsymbol{x}_{B,i}))(\boldsymbol{v}_2^\top \boldsymbol{f}_T^{(i)}(\boldsymbol{x}_{T,i})) \right)$$

$$\vdots$$

$$\le (2L)^{n-1} \mathcal{C}_{n,n-1} \left( \prod_{j=2}^{n-1} \mathcal{C}_{-j,-j} \right) \mathbb{E}_\epsilon \left( \sup_{(\boldsymbol{v}_1, \boldsymbol{v}_2) \in \mathcal{S}_n} \sum_{i=1}^m \epsilon_i (\boldsymbol{v}_1^\top \boldsymbol{x}_{B,i})(\boldsymbol{v}_2^\top \boldsymbol{x}_{T,i}) \right).$$

Using Lemma 6.4, the final bound on the empirical Rademacher complexity becomes,

$$\hat{\mathcal{R}}_m \le \frac{(2L)^{n-1} \mathcal{C}_{n,n-1}}{m} \left( \prod_{j=2}^{n-1} \mathcal{C}_{-j,-j} \right) \sqrt{\sum_{i=1}^m \| \tilde{\boldsymbol{x}}_i \|_2^2}.$$

Invoking the assumption that the input is bounded s.t $\mathbb{E}\left[ \| \tilde{\boldsymbol{x}}_B \|_2^2 \right] \le M_{x,B}^2$ and $\mathbb{E}\left[ \| \tilde{\boldsymbol{x}}_T \|_2^2 \right] \le M_{x,T}^2$, the average Rademacher complexity can be bounded as

$$\mathcal{R}_m \le \frac{(2L)^{n-1} \mathcal{C}_{n,n-1}}{\sqrt{m}} \left( \prod_{j=2}^{n-1} \mathcal{C}_{-j,-j} \right) M_{x,B} M_{x,T}.$$

$\square$

**Lemma 6.4.**

$$\mathbb{E}_\epsilon \left( \sup_{(\boldsymbol{v}_1, \boldsymbol{v}_2) \in \mathcal{S}_n} \sum_{i=1}^m \epsilon_i (\boldsymbol{v}_1^\top \boldsymbol{x}_{B,i})(\boldsymbol{v}_2^\top \boldsymbol{x}_{T,i}) \right) \le \sqrt{\sum_{i=1}^m \| \tilde{\boldsymbol{x}}_i \|_2^2}.$$

where $\tilde{\boldsymbol{x}}_i = \boldsymbol{x}_{B,i} \boldsymbol{x}_{T,i}^\top$ and $\mathcal{S}_n = S^{d_1-1} \times S^{d_2-1}$ and each $\epsilon_i \sim \pm 1$ uniformly.

Proof of the above lemma is given in Appendix D. In Appendix G we have setup a general framework for using Rademacher bounds as proven in the above theorem to prove generalization error bounds for DeepONets. And we will now invoke the result there to prove our main theorem about generalization error of DeepONets.

**Proof of Theorem 4.2**

*Proof.* We recall from Theorem 4.1 the following bound on Rademacher complexity for the given DeepONet class,

$$\mathcal{R}_m \leq \frac{(2L)^{n-1}\mathcal{C}_{n,n-1}}{\sqrt{m}} \left(\prod_{j=2}^{n-1} \mathcal{C}_{-j,-j}\right) M_{x,B} M_{x,T}$$

Combining the above with Proposition G.6 we have,

$$\mathbb{E}_{\{(f_i,\boldsymbol{x}_{T,i})\overset{iid}{\sim}\mathcal{D} \mid i=1...m\}} \left[\sup_{h\in\mathcal{H}}\left[\frac{1}{m}\sum_{i=1}^{m}\ell(h(f_i,\boldsymbol{x}_{T,i})) - \mathbb{E}_{(f,\boldsymbol{x}_T)\sim\mathcal{D}}\left[\ell(h(f,\boldsymbol{x}_T))\right]\right]\right]$$

$$\leq 2R \cdot \frac{(2L)^{n-1}\mathcal{C}_{n,n-1}}{\sqrt{m}} \left(\prod_{j=2}^{n-1} \mathcal{C}_{-j,-j}\right) M_{x,B} M_{x,T}$$

Note that Huber-loss $\ell_{H,\delta}$ is $\delta$-Lipshitz and further setting $\delta = \left(\frac{1}{2}\right)^{n-1}$ and rewriting the supremum over all $h \in \mathcal{H}$ as supremum over all $\boldsymbol{w} \in \mathcal{W}$ we can write the above inequality as

$$\mathbb{E}_{\{(f_i,\boldsymbol{x}_{T,i})\overset{iid}{\sim}\mathcal{D} \mid i=1...m\}} \left[\sup_{\boldsymbol{w}\in\mathcal{W}}\left[\frac{1}{m}\sum_{i=1}^{m}\ell_{H,\delta}(h(f_i,\boldsymbol{x}_{T,i})) - \mathbb{E}_{(f,\boldsymbol{x}_T)\sim\mathcal{D}}\left[\ell_{H,\delta}(h(f,\boldsymbol{x}_T))\right]\right]\right]$$

$$\leq \frac{(2L)^{n-1}\mathcal{C}_{n,n-1}}{2^{(n-2)}\sqrt{m}} \left(\prod_{j=2}^{n-1} \mathcal{C}_{-j,-j}\right) M_{x,B} M_{x,T}$$

Lastly, considering that the activation functions are $\sigma_1(x) = \sigma_2(x) = |x|$ which makes $L = 1$ in Lemma 6.1 we obtained the claimed bound,

$$\mathbb{E}_{\{(f_i,\boldsymbol{x}_{T,i})\overset{iid}{\sim}\mathcal{D} \mid i=1...m\}} \left[\sup_{\boldsymbol{w}\in\mathcal{W}}\left[\frac{1}{m}\sum_{i=1}^{m}\ell_{H,\delta}(h(f_i,\boldsymbol{x}_{T,i})) - \mathbb{E}_{(f,\boldsymbol{x}_T)\sim\mathcal{D}}\left[\ell_{H,\delta}(h(f,\boldsymbol{x}_T))\right]\right]\right]$$

$$\leq \frac{2\mathcal{C}_{n,n-1}}{\sqrt{m}} \left(\prod_{j=2}^{n-1} \mathcal{C}_{-j,-j}\right) M_{x,B} M_{x,T}$$

$\square$

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

## Appendix

## A   Evidence for Training DeepONets with $\tanh$ Activation and Huber Loss

Similar to the experiments in Section 4.3, in this section too we use DeepONets whose both branch and trunk networks are of depth of 3 and have a width of 100. However, unlike Section 4.3, the activation function used in the experiments displayed in Figure 4 is tanh. The chosen training loss is the same as in equation 11, where $\ell$ is set to the Huber loss with $\delta = \frac{1}{2^{3-1}} = 0.25$, the value as would be inspired by Theorem 4.2.

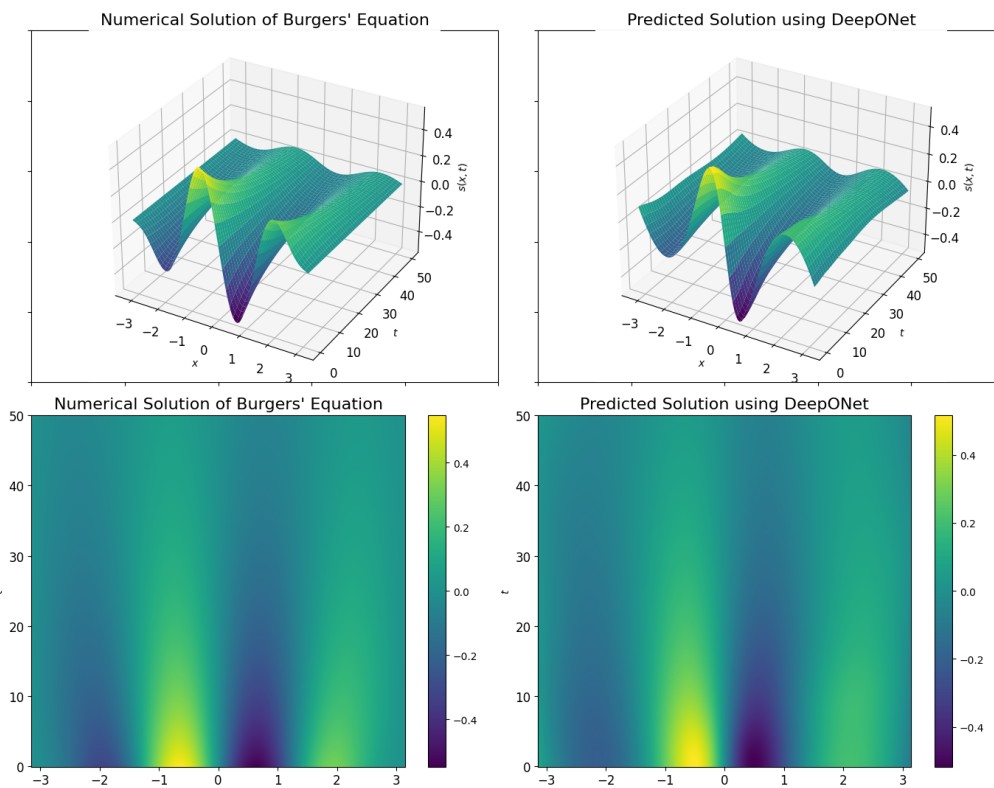

Figure 4: This plot demonstrates the ability of Huber loss trained DeepONets to predict the solution to a Burgers' P.D.E. on an arbitrarily chosen inhomogeneous term $u$, which is different from the $u$'s used for training the net.

The plots above motivate future studies exploring the possibility of Huber loss being good for training DeepONets.

## B   Empirical Study on the Behaviour of Generalization Error for Huber Loss at Different $\delta$

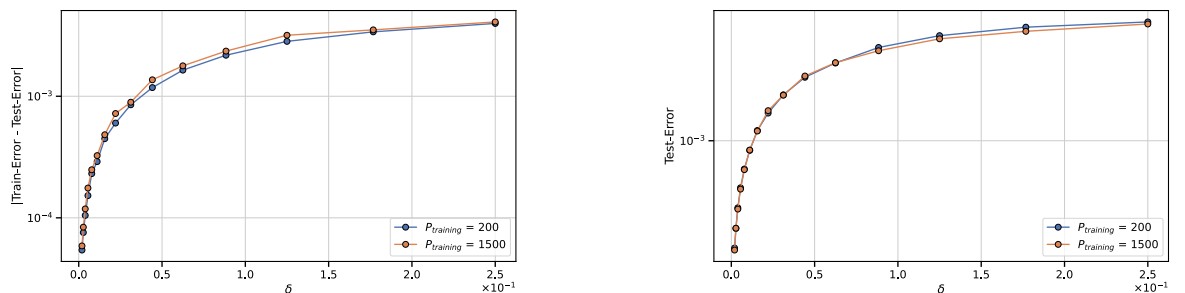

Figure 5: These plots demonstrate the behaviour of generalization error and test error for Huber loss trained DeepONets to predict the solution to a Burgers' P.D.E. at varying values of $\delta$ for 2 different dataset sizes.

Similar to the experiments in Section 4.3, in this section too we use DeepONets whose both branch and trunk networks are of depth 3 and have a width of 100. The selected training loss is identical to the one in Equation 11, with $\ell$ set to the Huber loss. The generalization error is plotted for various values of delta at $P_{\text{training}} = 200$ and 1500, while maintaining $N_{\text{training}} = 400$. In this setup, in Figure 5 we plot the generalization error for $\delta = \frac{1}{(\sqrt{2})^k}$ for $k = 4, 5, \cdots, 18$ to show how the generalization error as well as the test error drops as $\delta$ gets smaller.

## C  Review of the Universal Approximation Property of DeepONets

We follow the setup in Lanthaler et al. (2022) for a brief review of its key theorem that motivates DeepONets.

**Definition 7** (Solution Operator). Given $U \subset \mathbb{R}^n$ and $D \subset \mathbb{R}^d$, compact sets with boundaries, suppose we have a differential operator $\mathcal{L} : H^s(U) \to C(D)$, where $H^s(U)$ is the $L^2$-based Sobolev space for some $s > 0$. Denote the functions in the range of $\mathcal{L}$ to be "forcing" / "input" functions and those in the domain of $\mathcal{L}$ to be the "output" functions. Let $\partial U$ denote the boundary of $U$. Then given $g \in L^2(\partial U)$ and $f \in C(D)$ a solution to the differential system $(g, f, \mathcal{L})$ is a function $u^* \in H^s(U)$ s.t.

$$\mathcal{L}u^* = f \text{ s.t } u^* = g \text{ on } \partial U.$$

At a fixed "boundary condition" $g$ for such a differential system, we assume that the solutions for the above system for different forcing functions $f$ are given via an operator/map, $\mathcal{G}$ s.t.

$$\mathcal{G} : C(D) \to H^s(U) \text{ s.t } \mathcal{L} \circ \mathcal{G}(f) = f \text{ and } \mathcal{G}(f) = g \text{ on } \partial U.$$

Further, assume that $\mu$ is a probability measure on $C(D)$ s.t $\mathcal{G} \in L^2(\mu)$ i.e. $\int_{C(D)} \|\mathcal{G}(f)\|_{H^s(U)}^2 \, d\mu(f) < \infty$.

Multiple examples of $\mathcal{G}$ have been discussed in Lanthaler et al. (2022), and bounds on these operators evaluated – for instance, in Lemma 4.1 therein one can see the bounds on the $\mathcal{G}$ that corresponds to the the case of a forced pendulum that we used as a demonstrative example in Section 1.1

**Definition 8 (DeepONet (Version 2)).** Suppose $\forall f \in C(D), \boldsymbol{x}_B(f)$ is a discretization of $f$. Further suppose $\mathcal{A}$ is a branch net mapping to $\mathbb{R}^p$ and $\tau : \mathbb{R}^n \mapsto \mathbb{R}^{p+1}$ is a trunk net. Then a "DeepONet" ($\mathcal{N} : C(D) \mapsto L^2(U)$) is defined as,

$$\forall \boldsymbol{x}_T \in U \subset \mathbb{R}^n, \quad \mathcal{N}(f)(\boldsymbol{x}_T) := \tau_0(\boldsymbol{x}_T) + \sum_{k=1}^p \mathcal{A}_k(\boldsymbol{x}_B(f))\tau_k(\boldsymbol{x}_T).$$

**Theorem C.1 (Restatement of a Key Result from Lanthaler et al. (2022)).** Let $\mu$ be as in Definition 7. Let $\mathcal{G} : C(D) \to L^2(U)$ be a Borel measurable mapping, with $\mathcal{G} \in L^2(\mu)$, then for every $\epsilon > 0$, there exists an operator network $\mathcal{N} : C(D) \to L^2(U)$ as given above, such that

$$\|\mathcal{G} - \mathcal{N}\|_{L^2(\mu)} = \left( \int_{C(D)} \|\mathcal{G}(f) - \mathcal{N}(f)\|_{L^2(U)}^2 d\mu(f) \right)^{1/2} < \epsilon.$$

**Remark.** Henceforth $\boldsymbol{x}_B := \boldsymbol{x}_B(f)$ for any function $f$, and similarly $\boldsymbol{x}_{B,i}$ for a function $f_i$.

The above approximation guarantee between DeepONets ($\mathcal{N}$) and solution operators of differential equations ($\mathcal{G}$) clearly motivates the use of DeepONets for solving differential equations. In Deng et al. (2022), the authors present specific scenarios wherein particularly small DeepONets satisfy Theorem C.1. And as a counterpoint, in Mukherjee & Roy (2024), scenarios are demonstrated where small DeepONets would not be able to reduce the empirical training error below a certain threshold in the presence of noise in the data.

## D  Proof of Lemma 6.4

*Proof.* Denote by $\mathcal{S}_n = S^{d_1-1} \times S^{d_2-1}$. Define $\tilde{\mathbf{W}} = (\boldsymbol{v}_1\boldsymbol{v}_2^\top)$ and $\tilde{\boldsymbol{x}}_i = (\boldsymbol{x}_{B,i}(\boldsymbol{x}_{T,i})^\top)$. Then,

$$\mathbb{E}_\epsilon \left( \sup_{(\boldsymbol{v}_1, \boldsymbol{v}_2) \in \mathcal{S}_n} \sum_{i=1}^m \epsilon_i (\boldsymbol{v}_1^\top \boldsymbol{x}_{B,i})(\boldsymbol{v}_2^\top \boldsymbol{x}_{T,i}) \right) = \mathbb{E}_\epsilon \left( \sup_{(\boldsymbol{v}_1, \boldsymbol{v}_2) \in \mathcal{S}_n} \sum_{i=1}^m \epsilon_i \left( \sum_{k_1=1}^{d_1} v_{1,k_1} x_{B,i,k_1} \right) \left( \sum_{k_2=1}^{d_2} v_{2,k_2} x_{T,i,k_2} \right) \right)$$

$$= \mathbb{E}_\epsilon \left( \sup_{(\boldsymbol{v}_1, \boldsymbol{v}_2) \in \mathcal{S}_n} \sum_{i=1}^m \sum_{k_1=1}^{d_1} \sum_{k_2=1}^{d_2} \epsilon_i \left( v_{1,k_1} x_{B,i,k_1} \right) \left( v_{2,k_2} x_{T,i,k_2} \right) \right)$$

$$= \mathbb{E}_\epsilon \left( \sup_{(\boldsymbol{v}_1, \boldsymbol{v}_2) \in \mathcal{S}_n} \sum_{k_1=1}^{d_1} \sum_{k_2=1}^{d_2} v_{1,k_1} v_{2,k_2} \left( \sum_{i=1}^m \epsilon_i \left( x_{B,i,k_1} \right) \left( x_{T,i,k_2} \right) \right) \right)$$

Note that, $(\boldsymbol{v}_1, \boldsymbol{v}_2) \in \mathcal{S}_n \implies \left\| \tilde{\mathbf{W}} \right\|_2 = \sqrt{\sum_{k_1=1}^{d_1} \sum_{k_2=1}^{d_2} v_{1,k_1}^2 v_{2,k_2}^2} = \| \boldsymbol{v}_1 \|_2 \| \boldsymbol{v}_2 \|_2 = 1.$

Hence, we can enlarge the domain to get,

$$\leq \mathbb{E}_\epsilon \left( \sup_{\| \tilde{\mathbf{W}} \|_2 \leq 1} \sum_{k_1=1}^{d_1} \sum_{k_2=1}^{d_2} \tilde{W}_{(k_1,k_2)} \left( \sum_{i=1}^m \epsilon_i \, \tilde{x}_{i,k_1,k_2} \right) \right).$$

The above sum can be viewed as an inner product of 2 vectors in $\mathbb{R}^{d_1 \times d_2}$. We have,

$$= \mathbb{E}_\epsilon \left( \sup_{\| \tilde{\mathbf{W}} \|_2 \leq 1} \left\langle \tilde{\mathbf{W}}, \sum_{i=1}^m \epsilon_i \, \tilde{\boldsymbol{x}}_i \right\rangle \right) \leq \sqrt{\sum_{i=1}^m \| \tilde{\boldsymbol{x}}_i \|_2^2} \leq \sqrt{\sum_{i=1}^m \| \boldsymbol{x}_{B,i} \|_2^2 \| \boldsymbol{x}_{T,i} \|_2^2},$$

where the inequality follows from Theorem 5.5 in Ma (2021). $\qquad \square$

# E  Proof of Proposition 6.2

*Proof.* We recall from the setup of Definition 2 that $b_{-1}$ and $t_{-1}$ are the number of rows of the matrices $\boldsymbol{B}_{q_B-1}$ and $\boldsymbol{T}_{q_T-1}$ - and $b_{-2}$ and $t_{-2}$ are the output dimensions of the nets $\boldsymbol{f}'_B$ and $\boldsymbol{f}'_T$. From Definition 6 we further recall the definitions of the 2 sets of matrices $\mathcal{W}(\mathcal{C}_{q_B,q_T,q_B-1,q_T-1})$ and $\mathcal{W}_{\text{rest}}$ and we simplify the required empirical Rademacher complexity as,

$$m\hat{\mathcal{R}}_m = \mathbb{E}_\epsilon \left[ \sup_{(\mathcal{W}(\mathcal{C}_{q_B,q_T,q_B-1,q_T-1}),\mathcal{W}_{\text{rest}})} \sum_{i=1}^m \epsilon_i \left\langle \boldsymbol{f}_B(\boldsymbol{x}_{B,i}), \boldsymbol{f}_T(\boldsymbol{x}_{T,i}) \right\rangle \right]$$

$$= \mathbb{E}_\epsilon \left[ \sup_{(\mathcal{W}(\mathcal{C}_{q_B,q_T,q_B-1,q_T-1}),\mathcal{W}_{\text{rest}})} \left( \sum_{i=1}^m \epsilon_i \left\langle \boldsymbol{B}_{q_B} \sigma_1 \left( \boldsymbol{B}_{q_B-1} \boldsymbol{f}'_B(\boldsymbol{x}_{B,i}) \right), \boldsymbol{T}_{q_T} \sigma_2 \left( \boldsymbol{T}_{q_T-1} \boldsymbol{f}'_T(\boldsymbol{x}_{T,i}) \right) \right\rangle \right) \right]. \quad (17)$$

Here, we have substituted the definitions of the functions $\boldsymbol{f}'_B$ and the $\boldsymbol{f}'_T$ as given in the statement of Lemma 6.2. To ease notation now we define the following vectors,

$$\boldsymbol{b}_i \coloneqq \sigma_1 \left( \boldsymbol{B}_{q_B-1} \boldsymbol{f}'_B(\boldsymbol{x}_{B,i}) \right), \; \boldsymbol{t}_i \coloneqq \sigma_2 \left( \boldsymbol{T}_{q_T-1} \boldsymbol{f}'_T(\boldsymbol{x}_{T,i}) \right).$$

Then we can rewrite the above as,

$$
m\hat{\mathcal{R}}_m = \mathbb{E}_\epsilon\left[\sup_{(\mathcal{W}(\mathcal{C}_{q_B,q_T,q_B-1,q_T-1}),\mathcal{W}_{\text{rest}})}\left(\sum_{i=1}^m \epsilon_i \langle \boldsymbol{B}_{q_B}\boldsymbol{b}_i, \boldsymbol{T}_{q_T}\boldsymbol{t}_i\rangle\right)\right]
$$

$$
= \mathbb{E}_\epsilon\left[\sup_{(\mathcal{W}(\mathcal{C}_{q_B,q_T,q_B-1,q_T-1}),\mathcal{W}_{\text{rest}})}\left(\sum_{i=1}^m \epsilon_i \sum_{j=1}^p \boldsymbol{B}_{q_B,j}^\top \boldsymbol{b}_i \cdot \boldsymbol{T}_{q_T,j}^\top \boldsymbol{t}_i\right)\right]
$$

$$
= \mathbb{E}_\epsilon\left[\sup_{(\mathcal{W}(\mathcal{C}_{q_B,q_T,q_B-1,q_T-1}),\mathcal{W}_{\text{rest}})}\left(\sum_{i=1}^m \epsilon_i \sum_{j=1}^p \left(\sum_{k_1=1}^{b_{-1}} B_{q_B,j,k_1}b_{i,k_1}\right)\cdot\left(\sum_{k_2=1}^{t_{-1}} T_{q_T,j,k_2}t_{i,k_2}\right)\right)\right]
$$

$$
= \mathbb{E}_\epsilon\left[\sup_{(\mathcal{W}(\mathcal{C}_{q_B,q_T,q_B-1,q_T-1}),\mathcal{W}_{\text{rest}})}\left(\sum_{j=1}^p\sum_{k_1=1}^{b_{-1}}\sum_{k_2=1}^{t_{-1}} B_{q_B,j,k_1}T_{q_T,j,k_2}\sum_{i=1}^m \epsilon_i b_{i,k_1}\cdot t_{i,k_2}\right)\right]
$$

$$
= \mathbb{E}_\epsilon\left[\sup_{(\mathcal{W}(\mathcal{C}_{q_B,q_T,q_B-1,q_T-1}),\mathcal{W}_{\text{rest}})}\left(\sum_{j=1}^p\sum_{k_1=1}^{b_{-1}}\sum_{k_2=1}^{t_{-1}} B_{q_B,j,k_1}T_{q_T,j,k_2}\sum_{i=1}^m \epsilon_i b_{i,k_1}\cdot t_{i,k_2}\right)\right]
$$

$$
= \mathbb{E}_\epsilon\left[\sup_{\substack{\mathcal{W}(\mathcal{C}_{q_B,q_T,q_B-1,q_T-1})\\\mathcal{W}_{\text{rest}}}}\left(\sum_{j=1}^p\sum_{k_1=1}^{b_{-1}}\sum_{k_2=1}^{t_{-1}} (\boldsymbol{B}_{q_B,j}\boldsymbol{T}_{q_T,j}^\top)_{k_1,k_2}\right.\right.
$$
$$
\left.\left.\cdot\sum_{i=1}^m \epsilon_i\sigma_1\left(\boldsymbol{B}_{q_B-1}\boldsymbol{f}_B'(\boldsymbol{x}_{B,i})\right)_{k_1}\cdot\sigma_2\left(\boldsymbol{T}_{q_T-1}\boldsymbol{f}_T'(\boldsymbol{x}_{T,i})\right)_{k_2}\right)\right]
$$

$$
= \mathbb{E}_\epsilon\left[\sup_{\substack{\mathcal{W}(\mathcal{C}_{q_B,q_T,q_B-1,q_T-1})\\\mathcal{W}_{\text{rest}}}}\left(\sum_{j=1}^p\sum_{k_1=1}^{b_{-1}}\sum_{k_2=1}^{t_{-1}} (\boldsymbol{B}_{q_B,j}\boldsymbol{T}_{q_T,j}^\top)_{k_1,k_2}\cdot\|\boldsymbol{B}_{q_B-1,k_1}\|\cdot\|\boldsymbol{T}_{q_T-1,k_2}\|\right.\right.
$$
$$
\left.\left.\cdot\sum_{i=1}^m \epsilon_i\sigma_1\left(\hat{\boldsymbol{B}}_{q_B-1,k_1}^\top\boldsymbol{f}_B'(\boldsymbol{x}_{B,i})\right)\cdot\sigma_2\left(\hat{\boldsymbol{T}}_{q_T-1,k_2}^\top\boldsymbol{f}_T'(\boldsymbol{x}_{T,i})\right)\right)\right].
$$

In the last line above we have invoked the positive homogeneity of $\sigma_1$ and $\sigma_2$.

Now assume 2 vectors $\boldsymbol{v}$ and $\boldsymbol{w}$ of dimensions $b_{-1}-1\times t_{-1}-1$ s.t they are indexed by the tuple $(k_1,k_2)$ for $k_1\in\{1,\ldots,-1+b_{-1}\}$ and $k_2\in\{1,\ldots,-1+t_{-1}\}$ as follows,

$$
v_{(k_1,k_2)} \coloneqq \|\boldsymbol{B}_{q_B-1,k_1}\|\cdot\|\boldsymbol{T}_{q_T-1,k_2}\|\cdot\sum_{j=1}^p(\boldsymbol{B}_{q_B,j}\boldsymbol{T}_{q_T,j}^\top)_{k_1,k_2},
$$

$$
w_{(k_1,k_2)} \coloneqq \sum_{i=1}^m \epsilon_i\sigma_1\left(\hat{\boldsymbol{B}}_{q_B-1,k_1}^\top\boldsymbol{f}_B'(\boldsymbol{x}_{B,i})\right)\cdot\sigma_2\left(\hat{\boldsymbol{T}}_{q_T-1,k_2}^\top\boldsymbol{f}_T'(\boldsymbol{x}_{T,i})\right).
$$

Then we have,

$$
m\hat{\mathcal{R}}_m = \mathbb{E}_\epsilon\left[\sup_{(\mathcal{W}(\mathcal{C}_{q_B,q_T,q_B-1,q_T-1}),\mathcal{W}_{\text{rest}})}\left(\sum_{k_1=1}^{b_{-1}}\sum_{k_2=1}^{t_{-1}} v_{(k_1,k_2)}\cdot w_{(k_1,k_2)}\right)\right].
$$

Note that for any $\alpha,\beta\in\mathbb{R}^k$ we have, $\langle\alpha,\beta\rangle\le(\max_{p\in 1,\ldots,k}|\beta_p|)\cdot\sum_{j=1}^k|\alpha_j|$

$$
\le \mathbb{E}_\epsilon\left[\sup_{(\mathcal{W}(\mathcal{C}_{q_B,q_T,q_B-1,q_T-1}),\mathcal{W}_{\text{rest}})}\left(\sum_{k_1=1}^{b_{-1}}\sum_{k_2=1}^{t_{-1}} |v_{(k_1,k_2)}|\cdot\max_{\substack{k_1=1,\ldots,b_{-1}-1\\k_2=1,\ldots,t_{-1}-1}}|w_{(k_1,k_2)}|\right)\right].
$$

Now we define a set of weights $\mathcal{W}_{\text{except}-\text{outer}}$ which is an union of all possible weight matrices that are allowed in $\mathcal{W}_{\text{rest}}$ and all possible choices of $(\boldsymbol{B}_{q_B-1},\boldsymbol{T}_{q_T-1})$ that are allowed in the set $\mathcal{W}(\mathcal{C}_{q_B,q_T,q_B-1,q_T-1})$.

Then we recall the definition of $\mathcal{W}(\mathcal{C}_{q_B,q_T,q_B-1,q_T-1})$, to get,

$$m\hat{\mathcal{R}}_m \leq \mathbb{E}_\epsilon\left[\sup_{\mathcal{W}_{\text{except-outer}}}\left(\mathcal{C}_{q_B,q_T,q_B-1,q_T-1}\cdot\max_{\substack{k_1=1,\ldots,b_{-1}-1\\k_2=1,\ldots,t_{-1}-1}}\left|\sum_{i=1}^m\epsilon_i\sigma_1\left(\hat{\boldsymbol{B}}_{q_B-1,k_1}^\top\boldsymbol{f}_B'(\boldsymbol{x}_{B,i})\right)\cdot\sigma_2\left(\hat{\boldsymbol{T}}_{q_T-1,k_2}^\top\boldsymbol{f}_T'(\boldsymbol{x}_{T,i})\right)\right|\right)\right].$$

where

$$\mathcal{C}_{q_B,q_T,q_B-1,q_T-1} = \sum_{k_1=1}^{b_{-1}}\sum_{k_2=1}^{t_{-1}}\left|v_{(k_1,k_2)}\right|$$

$$= \sum_{k_1=1}^{b_{-1}}\sum_{k_2=1}^{t_{-1}}\|\boldsymbol{B}_{q_B-1,k_1}\|\cdot\|\boldsymbol{T}_{q_T-1,k_2}\|\left|\left[\sum_{j=1}^p\boldsymbol{B}_{q_B,j}\boldsymbol{T}_{q_T,j}^\top\right]_{k_1,k_2}\right|$$

Note that any pair of row directions in the pair of matrices $(\boldsymbol{B}_{q_B-1},\boldsymbol{T}_{q_T-1})$ is in $\in S^{b_{-2}-1}\times S^{t_{-2}-1}$. So a sup over the set of $\boldsymbol{B}_{q_B-1}$ and $\boldsymbol{T}_{q_T-1}$ that is allowed by the constraint of $\mathcal{C}_{q_B,q_T,q_B-1,q_T-1}$ and a subsequent max over the pairs of row directions can be upperbounded by a single sup over $S^{b_{-2}-1}\times S^{t_{-2}-1}$. Thus we have,

$$m\hat{\mathcal{R}}_m \leq \mathbb{E}_\epsilon\left[\sup_{\substack{(\boldsymbol{v},\boldsymbol{w})\in S^{b_{-2}-1}\times S^{t_{-2}-1}\\\mathcal{W}_{\text{rest}}}}\left(\mathcal{C}_{q_B,q_T,q_B-1,q_T-1}\cdot\left|\sum_{i=1}^m\epsilon_i\sigma_1\left(\boldsymbol{v}^\top\boldsymbol{f}_B'(\boldsymbol{x}_{B,i})\right)\cdot\sigma_2\left(\boldsymbol{w}^\top\boldsymbol{f}_T'(\boldsymbol{x}_{T,i})\right)\right|\right)\right].$$

Invoking Lemma H.1, we have

$$m\hat{\mathcal{R}}_m \leq \mathbb{E}_\epsilon\left[2\times\sup_{\substack{(\boldsymbol{v},\boldsymbol{w})\in S^{b_{-2}-1}\times S^{t_{-2}-1}\\\mathcal{W}_{\text{rest}}}}\left(\mathcal{C}_{q_B,q_T,q_B-1,q_T-1}\cdot\sum_{i=1}^m\epsilon_i\sigma_1\left(\boldsymbol{v}^\top\boldsymbol{f}_B'(\boldsymbol{x}_{B,i})\right)\cdot\sigma_2\left(\boldsymbol{w}^\top\boldsymbol{f}_T'(\boldsymbol{x}_{T,i})\right)\right)\right].$$

Now we invoke Lemma 6.1, assuming that Assumption 1 holds for $\sigma_1$ and $\sigma_2$ for some $L>0$, to get,

$$m\hat{\mathcal{R}}_m \leq 2L\cdot\mathcal{C}_{q_B,q_T,q_B-1,q_T-1}\cdot\mathbb{E}_\epsilon\left[\sup_{\substack{(\boldsymbol{v},\boldsymbol{w})\in S^{b_{-2}-1}\times S^{t_{-2}-1}\\\mathcal{W}_{\text{rest}}}}\left(\sum_{i=1}^m\epsilon_i\boldsymbol{v}^\top\boldsymbol{f}_B'(\boldsymbol{x}_{B,i})\cdot\boldsymbol{w}^\top\boldsymbol{f}_T'(\boldsymbol{x}_{T,i})\right)\right].$$

The above inequality is exactly what we set out to prove. $\qquad\square$

## F  Proof of Proposition 6.3

*Proof.* We start with the expression in the R.H.S. of the Proposition 6.2 and simplify it similarly as was done in its proof in the previous appendix. Denote $\mathcal{S}_2 = S^{b_{-2}-1}\times S^{t_{-2}-1}$.

$$\mathbb{E}_\epsilon\left[\sup_{\substack{(\boldsymbol{v},\boldsymbol{w})\in\mathcal{S}_2\\\mathcal{W}_{\text{rest}}}}\left(\sum_{i=1}^m\epsilon_i\boldsymbol{v}^\top\boldsymbol{f}_B'(\boldsymbol{x}_{B,i})\cdot\boldsymbol{w}^\top\boldsymbol{f}_T'(\boldsymbol{x}_{T,i})\right)\right]$$

$$=\mathbb{E}_\epsilon\left[\sup_{\substack{(\boldsymbol{v},\boldsymbol{w})\in\mathcal{S}_2\\\mathcal{W}_{\text{rest}}}}\left(\sum_{i=1}^m\epsilon_i\boldsymbol{v}^\top\sigma_1(\boldsymbol{B}_{q_B-2}\boldsymbol{f}_B''(\boldsymbol{x}_{B,i}))\cdot\boldsymbol{w}^\top\sigma_2(\boldsymbol{T}_{q_T-2}\boldsymbol{f}_T''(\boldsymbol{x}_{T,i}))\right)\right]$$

$$=\mathbb{E}_\epsilon\left[\sup_{\substack{(\boldsymbol{v},\boldsymbol{w})\in\mathcal{S}_2\\\mathcal{W}_{\text{rest}}}}\sum_{j_1=1}^{b_{-2}}\sum_{j_2=1}^{t_{-2}}(\boldsymbol{v}\boldsymbol{w}^\top)_{j_1,j_2}\|\boldsymbol{B}_{q_B-2,j_1}\|\|\boldsymbol{T}_{q_T-2,j_2}\|\cdot\sum_{i=1}^m\epsilon_i\sigma_1(\hat{\boldsymbol{B}}_{q_B-2,j_1}^\top\boldsymbol{f}_B''(\boldsymbol{x}_{B,i}))\sigma_2(\hat{\boldsymbol{T}}_{q_T-2,j_2}^\top\boldsymbol{f}_T''(\boldsymbol{x}_{T,i}))\right].$$

Define

$$\tilde{v}_{j_1,j_2} \coloneqq (\boldsymbol{v}\boldsymbol{w}^\top)_{j_1,j_2}\|\boldsymbol{B}_{q_B-2,j_1}\|\|\boldsymbol{T}_{q_T-2,j_2}\|$$

$$\tilde{w}_{j_1,j_2} \coloneqq \sum_{i=1}^m\epsilon_i\sigma_1(\hat{\boldsymbol{B}}_{q_B-2,j_1}^\top\boldsymbol{f}_B''(\boldsymbol{x}_{B,i}))\sigma_2(\hat{\boldsymbol{T}}_{q_T-2,j_2}^\top\boldsymbol{f}_T''(\boldsymbol{x}_{T,i})).$$

Thus we get,

$$\mathbb{E}_\epsilon\left[\sup_{\substack{(\boldsymbol{v},\boldsymbol{w})\in\mathcal{S}_2\\\mathcal{W}_{\mathrm{rest}}}}\left(\sum_{i=1}^m \epsilon_i \boldsymbol{v}^\top \boldsymbol{f}'_B(\boldsymbol{x}_{B,i})\cdot \boldsymbol{w}^\top \boldsymbol{f}'_T(\boldsymbol{x}_{T,i})\right)\right] = \mathbb{E}_\epsilon\left[\sup_{\substack{(\boldsymbol{v},\boldsymbol{w})\in\mathcal{S}_2\\\mathcal{W}_{\mathrm{rest}}}}\sum_{j_1=1}^{b_{-2}}\sum_{j_2=1}^{t_{-2}}\tilde{v}_{j_1,j_2}\tilde{w}_{j_1,j_2}\right]$$

$$\le \mathbb{E}_\epsilon\left[\sup_{\substack{(\boldsymbol{v},\boldsymbol{w})\in\mathcal{S}_2\\\mathcal{W}_{\mathrm{rest}}}}\left(\sum_{j_1=1}^{b_{-2}}\sum_{j_2=1}^{t_{-2}}|\tilde{v}_{j_1,j_2}|\cdot\max_{j_1,j_2}|\tilde{w}_{j_1,j_2}|\right)\right].$$

For ease of notation we define the set $\mathcal{W}_{-2,-2}$ as the set of matrices allowed in $\mathcal{W}_{rest}$ but with those with indices $q_B-2$ and $q_T-2$ additionally satisfying the constraint in equation 16 in the statement of the lemma.

$$\mathbb{E}_\epsilon\left[\sup_{\substack{(\boldsymbol{v},\boldsymbol{w})\in\mathcal{S}_2\\\mathcal{W}_{\mathrm{rest}}}}\left(\sum_{i=1}^m \epsilon_i \boldsymbol{v}^\top \boldsymbol{f}'_B(\boldsymbol{x}_{B,i})\cdot \boldsymbol{w}^\top \boldsymbol{f}'_T(\boldsymbol{x}_{T,i})\right)\right]$$

$$\le \mathcal{C}_{-2,-2}\mathbb{E}_\epsilon\left[\sup_{\mathcal{W}_{-2,-2}}\left(\max_{\substack{j_1=1,\ldots q_B-2\\j_2=1,\ldots,q_T-2}}|\tilde{w}_{j_1,j_2}|\right)\right]$$

$$\le \mathcal{C}_{-2,-2}\mathbb{E}_\epsilon\left(\sup_{\mathcal{W}_{-2,-2}}\left(\max_{\substack{j_1=1,\ldots q_B-2\\j_2=1,\ldots,q_T-2}}\left|\sum_{i=1}^m \epsilon_i \sigma_1(\hat{\boldsymbol{B}}^\top_{q_B-2,j_1}\boldsymbol{f}''_B(\boldsymbol{x}_{B,i}))\sigma_2(\hat{\boldsymbol{T}}^\top_{q_T-2,j_2}\boldsymbol{f}''_T(\boldsymbol{x}_{T,i}))\right|\right)\right).$$

We note that any pair of row directions in the pair of matrices $(\boldsymbol{B}_{q_B-2},\boldsymbol{T}_{q_T-2})$ is $\in S^{b_{-3}-1}\times S^{t_{-3}-1}$. So a sup over the set of $\boldsymbol{B}_{q_B-2}$ and $\boldsymbol{T}_{q_T-2}$ that is allowed by the constraint of $\mathcal{C}_{-2,-2}$ and a subsequent max over the pairs of row directions can be upper-bounded by a single sup over $S^{b_{-3}-1}\times S^{t_{-3}-1}$.

We recall the definition of $\mathcal{W}'_{rest}$ given in the lemma to conclude, ($\mathcal{S}_3 = S^{b_{-3}-1}\times S^{t_{-3}-1}$)

$$\mathbb{E}_\epsilon\left[\sup_{\substack{(\boldsymbol{v},\boldsymbol{w})\in\mathcal{S}_2\\\mathcal{W}_{\mathrm{rest}}}}\left(\sum_{i=1}^m \epsilon_i \boldsymbol{v}^\top \boldsymbol{f}'_B(\boldsymbol{x}_{B,i})\cdot \boldsymbol{w}^\top \boldsymbol{f}'_T(\boldsymbol{x}_{T,i})\right)\right]$$

$$\le \mathcal{C}_{-2,-2}\mathbb{E}_\epsilon\left(\sup_{\substack{(\boldsymbol{v},\boldsymbol{w})\in\mathcal{S}_3\\\mathcal{W}'_{rest}}}\left|\sum_{i=1}^m \epsilon_i \sigma_1(\boldsymbol{v}^\top \boldsymbol{f}''_B(\boldsymbol{x}_{B,i}))\sigma_2(\boldsymbol{w}^\top \boldsymbol{f}''_T(\boldsymbol{x}_{T,i}))\right|\right).$$

Invoking Lemma H.1,

$$\le 2\mathcal{C}_{-2,-2}\mathbb{E}_\epsilon\left(\sup_{\substack{(\boldsymbol{v},\boldsymbol{w})\in\mathcal{S}_3\\\mathcal{W}'_{rest}}}\sum_{i=1}^m \epsilon_i \sigma_1(\boldsymbol{v}^\top \boldsymbol{f}''_B(\boldsymbol{x}_{B,i}))\sigma_2(\boldsymbol{w}^\top \boldsymbol{f}''_T(\boldsymbol{x}_{T,i}))\right)$$

$$\le 2L\mathcal{C}_{-2,-2}\mathbb{E}_\epsilon\left(\sup_{\substack{(\boldsymbol{v},\boldsymbol{w})\in\mathcal{S}_3\\\mathcal{W}'_{rest}}}\sum_{i=1}^m \epsilon_i (\boldsymbol{v}^\top \boldsymbol{f}''_B(\boldsymbol{x}_{B,i}))(\boldsymbol{w}^\top \boldsymbol{f}''_T(\boldsymbol{x}_{T,i}))\right).$$

In the last step above we have invoked Lemma 6.1 using the fact that Assumption 1 is true about $\sigma_1$ and $\sigma_2$. $\qquad\square$

## G   Generalization Bound For DeepONets For Unbounded Data

**Definition 9** ($\phi_\mathcal{H}$ Loss Function Class)**.** Given $\mathcal{H}$ as defined in Theorem 4.2, we define the class $\phi_\mathcal{H}$ associated with $\mathcal{H}$ as,

$$\phi_\mathcal{H} := \{\mathcal{F}\times\mathbb{R}^{d_2}\ni (f,\mathbf{x}_T)\mapsto \ell(h(f,\mathbf{x}_T))\in[0,\infty)\mid h\in\mathcal{H}\}$$

where $\ell$ is some non-negative univariate (loss) function

Applying Theorem 4.13 of Ma (2021) to the above loss class we get the following theorem

**Theorem G.1.** Given $\phi_{\mathcal{H}}$ as in Definition 9 we have,

$$\mathbb{E}_{\{(f_i, \boldsymbol{x}_{T,i}) \overset{iid}{\sim} \mathcal{D} | i = 1 \ldots m\}} \left[ \sup_{h \in \mathcal{H}} \left[ \frac{1}{m} \sum_{i=1}^{m} \ell(h(f_i, \boldsymbol{x}_{T,i})) - \mathbb{E}_{(f, \boldsymbol{x}_T) \sim \mathcal{D}} \left[ \ell(h(f, \boldsymbol{x}_T)) \right] \right] \right] \leq 2\mathcal{R}_m(\phi_{\mathcal{H}})$$

where $m$ is the number of samples used in the empirical risk, $\mathcal{D}$ is a distribution over $\mathcal{F} \times \mathbb{R}^{d_2}$, and $\mathcal{R}_m$ is Rademacher complexity as defined in Equation 9.

**Lemma G.2** (Lipschitz Composition in Rademacher Complexity). Assume that $\phi : \mathbb{R} \mapsto \mathbb{R}$ is a $L_\phi$-Lipschitz continuous function, such that $|\phi(t) - \phi(s)| \leq L_\phi |t - s|$. Suppose $\mathcal{V} \subset \mathbb{R}^m$. Then,

$$\mathbb{E}_\epsilon \left[ \sup_{f \in \mathcal{V}} \frac{1}{m} \sum_{i=1}^{m} \epsilon_i \phi(f_i) \right] \leq L_\phi \, \mathbb{E}_\epsilon \left[ \sup_{f \in \mathcal{V}} \frac{1}{m} \sum_{i=1}^{m} \epsilon_i f_i \right].$$

**Lemma G.3** ($\mathcal{R}(\ell(\mathcal{H}))$ in terms of $\mathcal{R}(\mathcal{H})$). Given $\mathcal{H}$ as defined in Theorem 4.2, if $\ell(\cdot)$ is $R$–Lipschitz,

$$\mathbb{E}_\epsilon \left[ \sup_{h \in \mathcal{H}} \frac{1}{m} \sum_{i=1}^{m} \epsilon_i \, \ell(h(f_i, \boldsymbol{x}_{T,i})) \right] = \mathcal{R}(\ell(\mathcal{H})) \leq R \cdot \mathcal{R}(\mathcal{H}) = R \cdot \mathbb{E}_\epsilon \left[ \sup_{h \in \mathcal{H}} \frac{1}{m} \sum_{i=1}^{m} \epsilon_i \, h(f_i, \boldsymbol{x}_{T,i}) \right].$$

*Proof.* Any function $h \in \mathcal{H}$ maps from $\mathcal{F} \times \mathbb{R}^{d_2} \mapsto \mathbb{R}$. Hence, training data of size $m$ for the DeepONet is of the form $\{(f_i, \boldsymbol{x}_{T,i}) \in \mathcal{F} \times \mathbb{R}^{d_2}, i = 1, 2, \ldots, m\}$.

Consider the set $\mathcal{K} = \{(h(f_1, \boldsymbol{x}_1), \ldots, h(f_m, \boldsymbol{x}_m)) \mid h \in \mathcal{H}\} \subset \mathbb{R}^m$. Then we can re-write,

$$\mathcal{R}(\ell(\mathcal{H})) = \mathbb{E}_\epsilon \left[ \sup_{k \in \mathcal{K}} \frac{1}{m} \sum_{i=1}^{m} \epsilon_i \, \ell(k_i) \right].$$

Invoking that the Lipschitz constant for $\ell(\cdot)$ is $R$, and Lemma G.2 we have,

$$\mathcal{R}(\ell(\mathcal{H})) \leq R \, \mathbb{E}_\epsilon \left[ \sup_{k \in \mathcal{K}} \frac{1}{m} \sum_{i=1}^{m} \epsilon_i \, k_i \right],$$

which we recognize as the inequality we set out to prove,

$$\mathcal{R}(\ell(\mathcal{H})) \leq R \cdot \mathcal{R}(\mathcal{H}).$$

$\square$

**Lemma G.4** ($\mathcal{R}(\mathcal{H})$ in terms of $\mathcal{R}(\text{DeepONet})$). Recall that $\mathcal{W}$ is the set of allowed weights in the function class $\mathcal{H}$ (as defined in Theorem 4.2). Corrresponding to it define the class DeepONet s.t DeepONet$_{\boldsymbol{w}} \in$ DeepONet denotes a DeepONet with weight $\boldsymbol{w} \in \mathcal{W}$. Then the following holds,

$$\mathcal{R}(\text{DeepONet}) \coloneqq \mathbb{E}_\epsilon \left[ \sup_{\boldsymbol{w} \in \mathcal{W}} \frac{1}{m} \sum_{i=1}^{m} \epsilon_i \, \text{DeepONet}_{\boldsymbol{w}}(f_i, \boldsymbol{x}_{T,i}) \right] = \mathcal{R}(\mathcal{H}).$$

*Proof.* Realizing that taking a sup over $h \in \mathcal{H}$ and $\boldsymbol{w} \in \mathcal{W}$ are equivalent, we get,

$$\mathbb{E}_\epsilon \left[ \sup_{h \in \mathcal{H}} \frac{1}{m} \sum_{i=1}^{m} \epsilon_i \, h(f_i, \boldsymbol{x}_{T,i}) \right] = \mathbb{E}_\epsilon \left[ \sup_{\boldsymbol{w} \in \mathcal{W}} \frac{1}{m} \sum_{i=1}^{m} \epsilon_i \, (\mathcal{G}(f_i, \boldsymbol{x}_{T,i}) - \text{DeepONet}_{\boldsymbol{w}}(f_i, \boldsymbol{x}_{T,i})) \right],$$

where we have used $h(f_i, \boldsymbol{x}_{T,i}) = \mathcal{G}(f_i, \boldsymbol{x}_{T,i}) - \text{DeepONet}_{\boldsymbol{w}}(f_i, \boldsymbol{x}_{T,i})$. We can write $\mathcal{R}(\mathcal{H})$ as

$$
\begin{aligned}
\mathcal{R}(\mathcal{H}) &= \mathbb{E}_\epsilon \left[ \sup_{\boldsymbol{w} \in \mathcal{W}} \frac{1}{m} \sum_{i=1}^m \epsilon_i \left( \mathcal{G}(f_i, \boldsymbol{x}_{T,i}) - \text{DeepONet}_{\boldsymbol{w}}(f_i, \boldsymbol{x}_{T,i}) \right) \right] \\
&= \mathbb{E}_\epsilon \left[ \sup_{\boldsymbol{w} \in \mathcal{W}} \left\{ \frac{1}{m} \sum_{i=1}^m \epsilon_i \, \mathcal{G}(f_i, \boldsymbol{x}_{T,i}) - \frac{1}{m} \sum_{i=1}^m \epsilon_i \, \text{DeepONet}_{\boldsymbol{w}}(f_i, \boldsymbol{x}_{T,i}) \right\} \right] \\
&= \mathbb{E}_\epsilon \left[ \frac{1}{m} \sum_{i=1}^m \epsilon_i \, \mathcal{G}(f_i, \boldsymbol{x}_{T,i}) + \sup_{\boldsymbol{w} \in \mathcal{W}} \frac{1}{m} \sum_{i=1}^m (-\epsilon_i) \, \text{DeepONet}_{\boldsymbol{w}}(f_i, \boldsymbol{x}_{T,i}) \right] \\
&= 0 + \mathbb{E}_\epsilon \left[ \sup_{\boldsymbol{w} \in \mathcal{W}} \frac{1}{m} \sum_{i=1}^m \epsilon_i \, \text{DeepONet}_{\boldsymbol{w}}(f_i, \boldsymbol{x}_{T,i}) \right] \\
&= \mathcal{R}(\text{DeepONet}).
\end{aligned}
$$

since, $\epsilon = -\epsilon$ in distribution. $\qquad \square$

Hence combining the above two lemmas we get,

**Lemma G.5** ($\mathcal{R}(\ell(\mathcal{H}))$ in terms of $\mathcal{R}(\text{DeepONet})$)**.** Let $\ell(\cdot)$ be $R$–Lipschitz. Then,

$$
\mathcal{R}(\ell(\mathcal{H})) \le R \cdot \mathcal{R}(\text{DeepONet}).
$$

**Proposition G.6** (**Generalization Error Bound for DeepONet**)**.** Given $\phi_{\mathcal{H}}$ as in Definition 9 and let $\ell(\cdot)$ be $R$–Lipschitz. Then we have the following generalization bound,

$$
\mathbb{E}_{\{(f_i, \boldsymbol{x}_{T,i})^{iid}_\sim \mathcal{D} \ |i=1\ldots m\}} \left[ \sup_{h \in \mathcal{H}} \left[ \frac{1}{m} \sum_{i=1}^m \ell(h(f_i, \boldsymbol{x}_{T,i})) - \mathbb{E}_{(f,\boldsymbol{x}_T) \sim \mathcal{D}} \left[ \ell(h(f, \boldsymbol{x}_T)) \right] \right] \right] \le 2R \cdot \mathcal{R}_m
$$

where $\mathcal{R}_m$ is as in Theorem 4.1, and $\mathcal{D}$ is a distribution over $\mathcal{F} \times \mathbb{R}_{d_2}$

*Proof.* We recall from Theorem G.1 that,

$$
\mathbb{E}_{\{(f_i, \boldsymbol{x}_{T,i})^{iid}_\sim \mathcal{D}|i=1\ldots m\}} \left[ \sup_{h \in \mathcal{H}} \left[ \frac{1}{m} \sum_{i=1}^m \ell(h(f_i, \boldsymbol{x}_{T,i})) - \mathbb{E}_{(f,\boldsymbol{x}_T) \sim \mathcal{D}} \left[ \ell(h(f, \boldsymbol{x}_T)) \right] \right] \right] \le 2\mathcal{R}_m(\phi_{\mathcal{H}})
$$

Now, invoking Lemma G.5 and using the fact that $\ell(\cdot)$ is $R$–Lipschitz we get the claimed inequality. $\qquad \square$

## H    Contraction Lemmas

**Lemma H.1.** (From Ma (2021)) Let $\epsilon \sim \text{Uniform}(\{1, -1\}^m)$ and suppose we have functions $\boldsymbol{f}_\theta$ and $g_\theta$ parameterized by $\theta$ s.t, $(\mathbb{R}^k)^m \ni \boldsymbol{x} \mapsto \boldsymbol{f}_\theta(\boldsymbol{x}) = (g_\theta(x_1), \ldots, g_\theta(x_m)) \in \mathbb{R}^m$. Suppose that for any $\epsilon \in \{\pm 1\}^m$, $\sup_\theta \langle \epsilon, \boldsymbol{f}_\theta(x) \rangle \ge 0$. Then,

$$
\mathbb{E}_\epsilon \left[ \sup_\theta |\langle \epsilon, \boldsymbol{f}_\theta(\boldsymbol{x}) \rangle| \right] \le 2\mathbb{E}_\epsilon \left[ \sup_\theta \langle \epsilon, \boldsymbol{f}_\theta(\boldsymbol{x}) \rangle \right].
$$

*Proof.* Letting $\phi$ be the ReLU function, the lemma's assumption implies that $\sup_\theta \phi(\langle \epsilon, \boldsymbol{f}_\theta(\boldsymbol{x}) \rangle) = \sup_\theta \langle \epsilon, \boldsymbol{f}_\theta(\boldsymbol{x}) \rangle$ for any $\epsilon \in \{\pm 1\}^n$. Observing that $|z| = \phi(z) + \phi(-z)$,

$$
\begin{aligned}
\sup_\theta |\langle \epsilon, \boldsymbol{f}_\theta(x) \rangle| &= \sup_\theta \left[ \phi(\langle \epsilon, \boldsymbol{f}_\theta(x) \rangle) + \phi(\langle -\epsilon, \boldsymbol{f}_\theta(x) \rangle) \right] \\
&\le \sup_\theta \phi(\langle \epsilon, \boldsymbol{f}_\theta(x) \rangle) + \sup_\theta \phi(\langle -\epsilon, \boldsymbol{f}_\theta(x) \rangle) \\
&= \sup_\theta \langle \epsilon, \boldsymbol{f}_\theta(x) \rangle + \sup_\theta \langle -\epsilon, \boldsymbol{f}_\theta(x) \rangle.
\end{aligned}
$$

Taking the expectation over $\epsilon$ (and noting that $\epsilon$ and $-\epsilon$ have the same distribution) we get the desired conclusion. $\qquad \square$

## H.1    Proof of Lemma 6.1

*Proof.*

$$\mathcal{R}_{\mathrm{m}}(\phi_P \circ \mathcal{P} \cdot \phi_Q \circ \mathcal{Q}) = \frac{1}{m} \mathbb{E}_{\epsilon \sim U(\{-1,1\}^m)} \left[ \sup_{(p,q) \in \mathcal{P} \times \mathcal{Q}} \sum_{i=1}^{m} \epsilon_i \cdot \phi_P(p(\boldsymbol{x}_i)) \phi_Q(q(\boldsymbol{y}_i)) \right].$$

We explicitly open up the expectation on $\epsilon_1$ to get,

$$= \frac{1}{2m} \cdot \mathbb{E}_{(\epsilon_2,\ldots,\epsilon_m) \sim U(\{-1,1\}^{m-1})} \left[ \sup_{f,q \in \mathcal{P} \times \mathcal{Q}} \left( \phi_P(p(\boldsymbol{x}_1)) \phi_Q(q(\boldsymbol{y}_1)) + \sum_{i=2}^{m} \epsilon_i \cdot \phi_P(p(\boldsymbol{x}_i) \phi_Q(q(\boldsymbol{y}_i))) \right) + \right.$$
$$\left. \sup_{f',q' \in \mathcal{P} \times \mathcal{Q}} \left( -\phi_P(p'(\boldsymbol{x}_1)) \phi_Q(q'(\boldsymbol{y}_1)) + \sum_{i=2}^{m} \epsilon_i \cdot \phi_P(p'(\boldsymbol{x}_i)) \phi_Q(q'(\boldsymbol{y}_i)) \right) \right]$$

$$\leq \frac{1}{2m} \cdot \mathbb{E}_{(\epsilon_2,\ldots,\epsilon_m)} \left[ \sup_{((p,q),(p',q')) \in (\mathcal{P} \times \mathcal{Q})^2} \left( |\phi_P(p(\boldsymbol{x}_1)) \phi_Q(q(\boldsymbol{y}_1)) - \phi_P(p'(\boldsymbol{x}_1)) \phi_Q(q'(\boldsymbol{y}_1))| \right. \right.$$
$$\left. \left. + \sum_{i=2}^{m} \epsilon_i \left( \phi_P(p(\boldsymbol{x}_i)) \phi_Q(q(\boldsymbol{y}_i)) + \phi_P(p'(\boldsymbol{x}_i)) \phi_Q(q'(\boldsymbol{y}_i)) \right) \right) \right]$$

By invoking the the assumption 1 we get,

$$\leq \frac{1}{2m} \cdot \mathbb{E}_{(\epsilon_2,\ldots,\epsilon_m)} \left[ \sup_{((p,q),(p',q')) \in (\mathcal{P} \times \mathcal{Q})^2} \left( L \cdot |p(\boldsymbol{x}_1)q(\boldsymbol{y}_1) - p'(\boldsymbol{x}_1)q'(\boldsymbol{y}_1)| \right. \right.$$
$$\left. \left. + \sum_{i=2}^{m} \epsilon_i \left( \phi_P(p(\boldsymbol{x}_i)) \phi_Q(q(\boldsymbol{y}_i)) + \phi_P(p'(\boldsymbol{x}_i)) \phi_Q(q'(\boldsymbol{y}_i)) \right) \right) \right]$$

Now we invoke the fact that inside a supremum, an absolute value function is redundant, since anyway the higher combination will get picked. Thus we can rearrange the above to get,

$$\leq \frac{1}{2m} \cdot \mathbb{E}_{(\epsilon_2,\ldots,\epsilon_m)} \left[ \sup_{(p,q) \in \mathcal{P} \times \mathcal{Q}} \left( L \cdot p(\boldsymbol{x}_1)q(\boldsymbol{y}_1) + \sum_{i=2}^{m} \epsilon_i \cdot \phi_P(p(\boldsymbol{x}_i)) \phi_Q(q(\boldsymbol{y}_i)) \right) \right.$$
$$\left. + \sup_{(p',q') \in \mathcal{P} \times \mathcal{Q}} \left( -L \cdot p'(\boldsymbol{x}_1)q'(\boldsymbol{y}_1) + \sum_{i=2}^{m} \epsilon_i \cdot \phi_P(p'(\boldsymbol{x}_i)) \phi_Q(q'(\boldsymbol{y}_i)) \right) \right]$$

$$\leq \frac{1}{m} \cdot \mathbb{E}_{\epsilon} \left[ \sup_{(p,q) \in \mathcal{P} \times \mathcal{Q}} \left( \epsilon_1 \cdot L \cdot p(\boldsymbol{x}_1)q(\boldsymbol{y}_1) + \sum_{i=2}^{m} \epsilon_i \cdot \phi_P(p(\boldsymbol{x}_i)) \phi_Q(q(\boldsymbol{y}_i)) \right) \right]$$

Re-iterating the above argument for $i = 2, 3, , \ldots, m$ we get,

$$= \frac{1}{m} \mathbb{E}_{\epsilon \sim U(\{-1,1\}^m)} \sup_{(p,q) \in \mathcal{P} \times \mathcal{Q}} \left[ \sum_{i=1}^{m} \epsilon_i L \cdot p(\boldsymbol{x}_i)q(\boldsymbol{y}_i) \right]$$
$$= L\mathcal{R}_{\mathrm{m}}(\mathcal{P} \cdot \mathcal{Q}).$$

$\square$

## H.2 A Contraction Lemma with Biased Absolute Functions

Suppose $\mathcal{F}$ is set of functions with a common domain $D$ and a common range space, and $B$ is a subset of that range space. Then we denote a new function space $\mathcal{F} + B \coloneqq \{x \mapsto f(x) + \boldsymbol{b}, \ \forall x \in D \mid f \in \mathcal{F}, \boldsymbol{b} \in B\}$.

**Lemma H.2.** Let $\mathcal{P}$ and $\mathcal{G}$ be set of functions valued in $\mathbb{R}$ closed under negation. Given points $\{\boldsymbol{x}_i \mid i = 1, \dots, m\}$, $\{\boldsymbol{y}_i \mid i = 1, \dots, m\}$ in the domain of the functions then we have the following inequality of Rademacher complexities - where both the sides are being evaluated on this same set of points,

$$\mathcal{R}_{\mathrm{m}}(|\mathcal{P} + B_1| \cdot |\mathcal{G} + B_2|) \le \mathcal{R}_{\mathrm{m}}(\mathcal{P} \cdot \mathcal{G}) + |B_2|\mathcal{R}_{\mathrm{m}}(\mathcal{P}) + |B_1|\mathcal{R}_{\mathrm{m}}(\mathcal{G})$$

*Proof.* From the definition of Rademacher complexity it follows that,

$$\mathcal{R}_{\mathrm{m}}(|\mathcal{P} + B_1| \cdot |\mathcal{Q} + B_2|) = \frac{1}{m}\mathbb{E}_{\epsilon \sim U(\{-1,1\}^m)} \sup_{p,q \in \mathcal{P} \times \mathcal{Q}} \left[\sum_{i=1}^m \epsilon_i \cdot |p(\boldsymbol{x}_i) + B_1||q(\boldsymbol{y}_i) + B_2|\right].$$

We explicitly open up the expectation on $\epsilon_1$ to get,

$$= \frac{1}{2m} \cdot \mathbb{E}_{(\epsilon_2, \dots, \epsilon_m) \sim U(\{-1,1\}^{m-1})} \left[ \sup_{p,q \in \mathcal{P} \times \mathcal{Q}} \left( |p(\boldsymbol{x}_1) + B_1||q(\boldsymbol{y}_1) + B_2| + \sum_{i=2}^m \epsilon_i \cdot \phi_P(p(\boldsymbol{x}_i))\phi_Q(q(\boldsymbol{y}_i)) \right) + \right.$$
$$\left. \sup_{p',q' \in \mathcal{P} \times \mathcal{Q}} \left( -|p'(\boldsymbol{x}_1) + B_1||q'(\boldsymbol{y}_1) + B_2| + \sum_{i=2}^m \epsilon_i \cdot \phi_P(p'(\boldsymbol{x}_i))\phi_Q(q'(\boldsymbol{y}_i)) \right) \right] \quad (18)$$

$$\le \frac{1}{2m} \cdot \mathbb{E}_{(\epsilon_2, \dots, \epsilon_m)} \left[ \sup_{p,p',q,q' \in \mathcal{P} \times \mathcal{P} \times \mathcal{Q} \times \mathcal{Q}} \left( |p(\boldsymbol{x}_1)q(\boldsymbol{y}_1) - p'(\boldsymbol{x}_1)q'(\boldsymbol{y}_1)| + |B_2||p(\boldsymbol{x}_1) - p'(\boldsymbol{x}_1)| + |B_1||q(\boldsymbol{y}_1) - q'(\boldsymbol{y}_1)| \right. \right.$$
$$\left. \left. + \sum_{i=2}^m \epsilon_i \left(\phi_P(p(\boldsymbol{x}_i))\phi_Q(q(\boldsymbol{y}_i)) + \phi_P(p'(\boldsymbol{x}_i))\phi_Q(q'(\boldsymbol{y}_i))\right) \right) \right] \quad (19)$$

$$= \frac{1}{2m} \cdot \mathbb{E}_{(\epsilon_2, \dots, \epsilon_m)} \left[ \sup_{p,q \in \mathcal{P} \times \mathcal{Q}} \left( p(\boldsymbol{x}_1)q(\boldsymbol{y}_1) + |B_2|p(\boldsymbol{x}_1) + |B_1|q(\boldsymbol{y}_1) + \sum_{i=2}^m \epsilon_i \cdot \phi_P(p(\boldsymbol{x}_i))\phi_Q(q(\boldsymbol{y}_i)) \right) \right.$$
$$\left. + \sup_{p',q' \in \mathcal{P} \times \mathcal{Q}} \left( -p'(\boldsymbol{x}_1)q'(\boldsymbol{y}_1) - |B_2|p'(\boldsymbol{x}_1) - |B_1|q'(\boldsymbol{y}_1) + \sum_{i=2}^m \epsilon_i \cdot \phi_P(p'(\boldsymbol{x}_i))\phi_Q(q'(\boldsymbol{y}_i)) \right) \right] \quad (20)$$

$$\le \frac{1}{m} \cdot \mathbb{E}_{\epsilon} \left[ \sup_{f,g \in \mathcal{P} \times \mathcal{Q}} \left( \epsilon_1 \cdot \left(p(\boldsymbol{x}_1)q(\boldsymbol{y}_1) + |B_2|p(\boldsymbol{x}_1) + |B_1|q(\boldsymbol{y}_1)\right) + \sum_{i=2}^m \epsilon_i \cdot \phi_P(p(\boldsymbol{x}_i))\phi_Q(q(\boldsymbol{y}_i)) \right) \right] \quad (21)$$

Iterating this, we get

$$\le \frac{1}{m}\mathbb{E}_{\epsilon \sim U(\{-1,1\}^m)} \sup_{p,q \in \mathcal{P} \times \mathcal{Q}} \left[ \sum_{i=1}^m \epsilon_i \cdot (p(\boldsymbol{x}_i)q(\boldsymbol{y}_i) + |B_2|p(\boldsymbol{x}_i) + |B_1|q(\boldsymbol{y}_i)) \right]$$
$$\le \mathcal{R}_{\mathrm{m}}(\mathcal{P} \cdot \mathcal{Q}) + |B_2|\mathcal{R}_{\mathrm{m}}(\mathcal{P}) + |B_1|\mathcal{R}_{\mathrm{m}}(\mathcal{Q}).$$

In equation 19, we have used the triangle inequality, followed by

$$\left| |p(\boldsymbol{x}_1) + B_1| \cdot |q(\boldsymbol{y}_1) + B_2| - |p'(\boldsymbol{x}_1) + B_1| \cdot |q'(\boldsymbol{y}_1) + B_2| \right| \le \left| (p(\boldsymbol{x}_1) + B_1)(q(\boldsymbol{y}_1) + B_2) - (p'(\boldsymbol{x}_1) + B_1)(q'(\boldsymbol{y}_1) + B_2) \right|$$

In equation 20, we could open up the $|\cdot|$ since the supremum would be positive. This follows from the fact that $\mathcal{P}$ and $\mathcal{Q}$ are closed under multiplication by $-1$, which implies $\sup_{f \in F} f = \sup_{f \in -F} f = \sup_{f \in F} -f = \sup_{f \in F} |f|$

$\square$

# I Converting ReLU–DeepONets to abs–DeepONets

Firstly, we recall that the map $\mathbb{R}^q \ni \boldsymbol{x} \mapsto \boldsymbol{x} \in \mathbb{R}^q$ is an exact depth 2 ReLU net – one which passes every coordinate of the input through the ReLU net $\mathbb{R} \ni z \mapsto \max\{0, z\} - \max\{0, -z\} \in \mathbb{R}$. Using this, given any ReLU DeepONet, we can symmetrize the depths between the branch and the trunk by attaching the required number of identity computing layers at the input to the shorter side.

Secondly, for typical DeepONet experiments, one can assume that the the set of all possible input data is bounded. Combining this with the assumption of boundedness of the allowed matrices, we conclude that $\exists \, \mathcal{B} > 0$ s.t the input to any ReLU gate in any DeepONet in the given class is bounded by $\mathcal{B}$. Now we observe that $\forall |z| \le \mathcal{B}$, we can rewrite the map $z \mapsto \max\{0, z\}$ as, $z \mapsto \frac{1}{2}|z| + \frac{1}{4}|z + \mathcal{B}| - \frac{1}{4}|z - \mathcal{B}|$.

Hence, doing the above replacement at every gate we can rewrite any ReLU DeepONet (without biases) as a DeepONet using only absolute value activations but with biases – and computing the same function on the same bounded domain, at the cost of increasing the size of the branch and the trunk net by a factor of 3. Thirdly, a similar result as in Lemma 6.1 continues to hold for this setup as given in Lemma H.2. Note that, Lemma H.2 bounds the Rademacher complexity of a DeepONet class with the branch and trunk depths $(k, k)$ by a linear sum of that of $(k - 1, k - 1)$ and sum of a $(k, 0)$ and a $(0, k)$ DeepONet. While the first term can be recursed on again (using identical techniques as used to prove Theorem 4.1), for the last two one can invoke Theorem 1 of Golowich et al. (2018).

Thus, we observe that following the same arguments as in the proof for Theorem 4.1 we can derive an analogous bound for arbitrary ReLU DeepONet classes - but with twice the depth of the DeepONet number of extra terms in the R.H.S. These extra terms would come in pairs – each pair consisting of a Rademacher complexity bound on a standard net, one from the branch side and one from the trunk side and of decreasing depths.

# J Choosing $\tilde{\mathcal{C}}_{-k,-k}$

Defining $\left[ X_{j_1, j_2} \right]_{\substack{j_1 = 1, \ldots, b_{-k} \\ k_1 = 1, \ldots, t_{-k}}} := \boldsymbol{X} \in \mathbb{R}^{b_{-k} \times t_{-k}}$ s.t $X_{j_1, j_2} := \|\boldsymbol{B}_{n-k, j_1}\| \cdot \|\boldsymbol{T}_{n-k, j_2}\|$, we can simplify as follows the expression defining $\mathcal{C}_{-k,-k}$ in equation 4,

$$\sup_{(\boldsymbol{v}, \boldsymbol{w}) \in S^{-1+b_{-k}} \times S^{-1+t_{-k}}} \sum_{j_1=1}^{b_{-k}} \sum_{j_2=1}^{t_{-k}} \left|(\boldsymbol{v}\boldsymbol{w}^\top)_{j_1, j_2}\right| \|\boldsymbol{B}_{n-k, j_1}\| \|\boldsymbol{T}_{n-k, j_2}\| = \sup_{(\boldsymbol{v}, \boldsymbol{w}) \in S^{-1+b_{-k}} \times S^{-1+t_{-k}}} \sum_{j_1=1}^{b_{-k}} \sum_{j_2=1}^{t_{-k}} |v_{j_1}| \|\boldsymbol{B}_{n-k, j_1}\| \|\boldsymbol{T}_{n-k, j_2}\| |w_{j_2}|$$

$$= \sup_{(\boldsymbol{v}, \boldsymbol{w}) \in S^{-1+b_{-k}} \times S^{-1+t_{-k}}} \sum_{j_1=1}^{b_{-k}} \sum_{j_2=1}^{t_{-k}} |v_{j_1}| \cdot \boldsymbol{X}_{j_1, j_2} \cdot |w_{j_2}| = \sup_{(\boldsymbol{v}, \boldsymbol{w}) \in S^{-1+b_{-k}} \times S^{-1+t_{-k}}} \tilde{\boldsymbol{v}}^\top \boldsymbol{X} \tilde{\boldsymbol{w}}$$

In the last line above we defined vectors $\tilde{\boldsymbol{v}}$ and $\tilde{\boldsymbol{w}}$ of appropriate dimensions s.t $\tilde{v}_i = |v_i|$ and $\tilde{w}_i = |w_i|$. Then note that, $\tilde{\mathbf{v}}^\top \mathbf{X} \tilde{\mathbf{w}} \le \|\tilde{\mathbf{v}}\|_2 \|\mathbf{X} \tilde{\mathbf{w}}\|_2 \le \|\tilde{\mathbf{v}}\|_2 \|\mathbf{X}\| \|\tilde{\mathbf{w}}\|_2$

In above $\|\mathbf{X}\|$ is the spectral norm of $\mathbf{X}$. Therefore we have the following upperbound,

$$\sup_{\boldsymbol{v}, \boldsymbol{w}} \|\tilde{\mathbf{v}}\|_2 \|\mathbf{X}\| \|\tilde{\mathbf{w}}\|_2 \le \sup_{\boldsymbol{v}} \|\tilde{\mathbf{v}}\|_2 \cdot \sup_{\boldsymbol{w}} \|\tilde{\mathbf{w}}\|_2 \cdot \|\mathbf{X}\|$$

Since $\|\tilde{\boldsymbol{v}}\| = \|\boldsymbol{v}\| \le 1$ and $\|\tilde{\boldsymbol{w}}\| = \|\boldsymbol{w}\| \le 1$ we have,

$$\sup_{(\boldsymbol{v}, \boldsymbol{w}) \in S^{-1+b_{-k}} \times S^{-1+t_{-k}}} \tilde{\boldsymbol{v}}^\top \boldsymbol{X} \tilde{\boldsymbol{w}} \le \left\| \left[ \|\boldsymbol{B}_{n-k, j_1}\| \cdot \|\boldsymbol{T}_{n-k, j_2}\| \right]_{\substack{j_1 = 1, \ldots, b_{-k} \\ k_1 = 1, \ldots, t_{-k}}} \right\|$$

Thus it follows that the RHS of the above inequality gives an intuitive candidate for the quantity $\tilde{\mathcal{C}}_{-k,-k}$.

## K    Behaviour of Rademacher Bound for $\ell_2$-loss

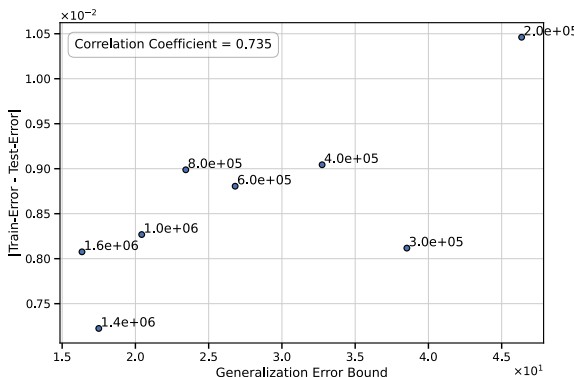

Figure 6: The above plot shows the behaviour of the measured generalization error with respect to $\frac{\mathcal{C}_{3,2}\,\tilde{\mathcal{C}}_{-2,-2}}{\sqrt{m}}$ for training DeepONets to solve Burgers' PDE using the empirical loss as given in equation 11, specialized to the $\ell_2$ loss and for the branch and the trunk nets being of depth 3. Each point is labelled by the number of training data used in that experiment.

## L    Link to Code

Here is the link to the GitHub repository containing our code for training a DeepONet for the Heat and Burgers' P.D.Es.

## M    Plot Showing the Evolution of Solution to the 2D Heat P.D.E. with Time

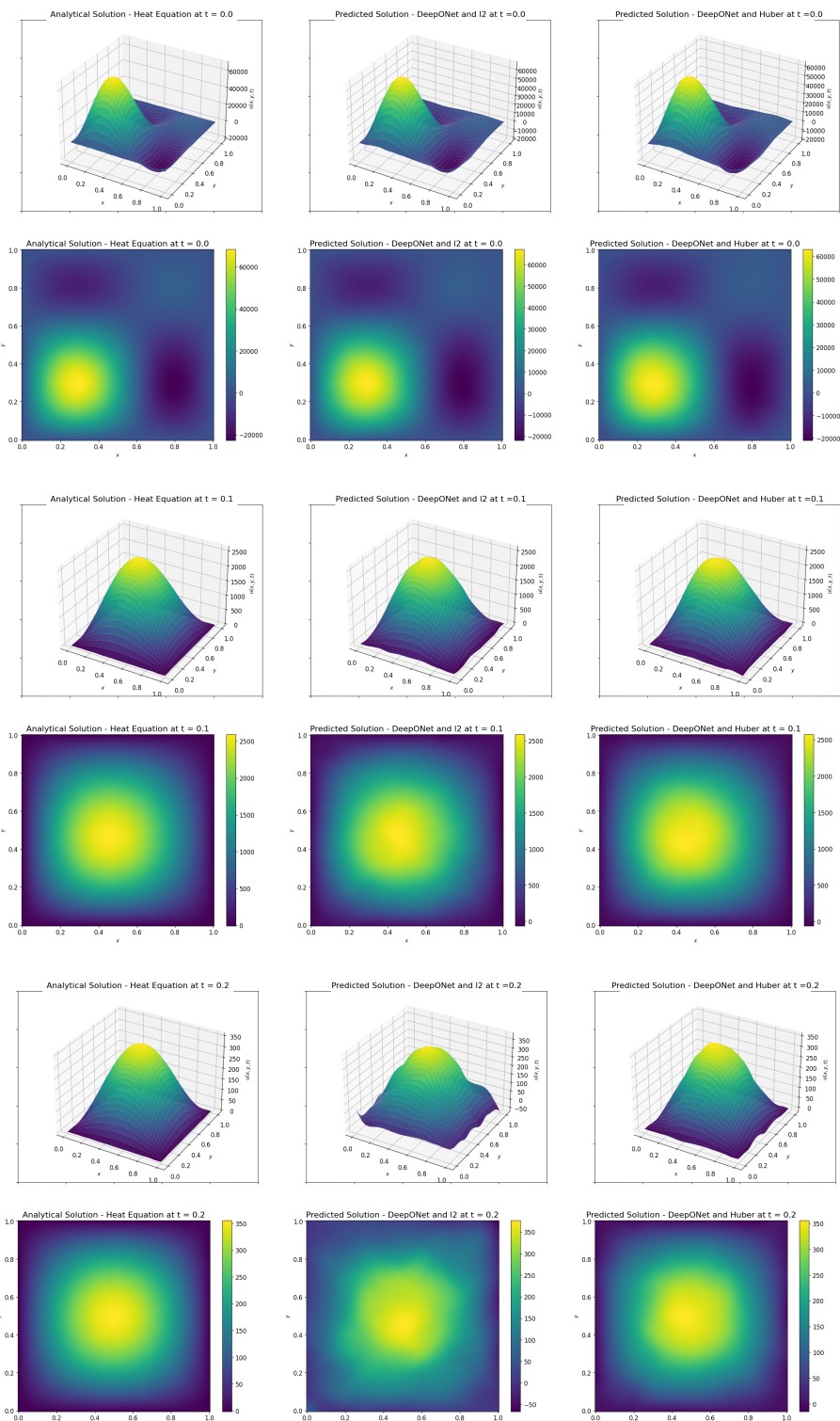

Figure 7: This plot demonstrates the solution to 2D Heat P.D.E. at different times. In each figure, we have shown the analytical solution first followed by the prediction from the DeepONet trained with $\ell_2$ and Huber loss at $\delta = 0.25$. Here, the DeepONets we used have ReLU activation functions and include bias terms.

