# OpenReview forum: "Towards Size-Independent Generalization Bounds for Deep Operator Nets"
_TMLR — Accepted by TMLR_

### Review · Reviewer_NK7K · 2024-06-19

**Summary Of Contributions:**

The paper provides a theoretical analysis of DeepONets, focusing on the generalization bound with two different loss functions: $l_2$ loss and Huber loss. The results surpass previous findings in several ways: 1. Their generalization bound is independent of the number of training parameters, explaining the success of overparameterized architectures in modern models. 2. They extended the results to Huber loss, demonstrating its empirical effectiveness in training neural networks to solve partial differential equations (PDEs). 3. Based on these findings, the paper potentially offers insights into how the choice of the hypothesis class, the choice of the loss function, and the data distribution interact to determine the ability to find optimal solutions through empirical estimates.

**Audience:**

Yes

**Broader Impact Concerns:**

No impact concerns.

**Claims And Evidence:**

Yes

**Requested Changes:**

Requested Changes:

1.Add the insights mentioned in the Introduction section and include experiments to validate them (related to Weakness 1).
2.Provide more discussion on their generalization bounds and assumptions.

**Strengths And Weaknesses:**

1.The paper is well-written and well-organized.
2.The results presented effectively explain the training of DeepONets. For instance, the model-size-independent results improve upon previous work and may provide valuable insights (see below weakness 1).
3.It discusses two distinct loss functions, $l_2$ loss and Huber loss, showcasing the extensibility of their work.
4.The experiments conducted are good, validating their theoretical claims.

Weaknesses:

1.The discussion on their insight (how the choice of the hypothesis class, the choice of the loss function, and the data distribution interact to determine the ability to find optimal solutions through empirical estimates) is missing, despite being mentioned in the Introduction section.

2.The explanation of their theoretical results is limited. The authors should elaborate on why the results are independent of widths, depths, and parameters. Are these terms synonymous? Could the summation of rows of the weight matrices be considered one of the parameters?

3.Assumption 1 appears strong. Are there related works that use the same assumption? Please provide references. Furthermore, if the activation function satisfies Assumption 1, are there any other conditions it must meet, such as monotonicity?

---

> ### Author Response · Authors · 2024-10-17
> **Responses to the Comments**
>
> We thank you for your careful reading of our draft and your detailed comments. We have implemented as many of your suggestions and below we give our detailed responses to the main queries.
>
> > 1. Add the insights mentioned in the Introduction section and include experiments to validate them (related to Weakness 1).
>
> In addition to our previous experiments on the Burgers' PDE, we have now included experiments using the Huber loss on the Heat PDE in Section 4.3.2. In both cases, we demonstrate that the Rademacher bound under Huber loss exhibits a strong correlation with the generalization error across varying amounts of training data, for two distinct Huber delta values. In the newly added Appendix M we have shown that even beyond the ambit of our assumptions, DeepONet trained with Huber loss performs better than $l_2$ loss.
>
> > 2. Provide more discussion on their generalization bounds and assumptions.
>
> Please note that in the introduction, paragraph 8, we have now added a discussion on how Rademacher complexity leads to sample complexity estimates. This relates to our earlier mention of the interaction between the hypothesis class, choice of loss function, and data distribution in determining the ability to find low-risk predictors using empirical estimates.
>
> In Appendix I, we discuss Assumption 1 further and explain how training a DeepONet with ReLU activation, but without biases, can be encompassed by the current setup.
>
> >3. Could the summation of rows of the weight matrices be considered one of the parameters?
>
> We suggest that the bounds should be read by first choosing the function space of which the Rademacher complexity is being computed and this function space of DeepONets, at any fixed depth, is being specified as that subset whose weights satisfy equation 4. Hence, the dependency of the Rademacher upperbound on all $\mathcal{C}$ terms should be read as a constant, and this way we conclude that the bound is not explicitly dependent on the width and hence the number of trainable parameters of the neural net. We note that this interpretation is consistent with the usual conventions of generalization-bound theory of neural nets - as reviewed in Section 2. Thus we can say that thus we have obtained size-independent generalization bounds - even for DeepONets and also while not making any assumptions on the target operator $\mathcal{G}$.

---

### Review · Reviewer_mR3e · 2024-06-28

**Summary Of Contributions:**

This work shows Rademacher complexity of DeepONets which scales well by applying “peeling” lemma and a variation of the Talagrand contraction lemma.  And it suggests the use of Huber type loss.  Some of the theoretical claims are reflected by toy experimental examples.

**Audience:**

Yes

**Claims And Evidence:**

Yes

**Requested Changes:**

1. There are several comparisons to existing work; please summarize them concisely in one section or so so that the relation can be clear.  Especially, I would like to see a subsection for related work on sample complexity and another subsection for related work on mathematical proof techniques.

2. It seems the authors separate meta summary of the work and mathematical setups, which looks good to me; however, if that is the case, meta summary part may need to be more intuitive so that there won’t be much overlap with the mathematical setup parts.

3. Experimental sections are not really well structured; please make it flow well and add sufficient descriptions to each figure.  Also the texts in figures are a bit too small.

4. It says the complexity does not (explicitly) depend on width and depth; how does the constants $C$ affect the complexity?  It seems exponential?

5. Definition 6 is a bit hard to parse.

6. page 28 bottom; “since the supremum would be positive…” can you elaborate more on this?

7. $f$ and $f'$ are introduced; it seems $f'$ is used differently in different parts. (like in page 26)  Can you unify them?

Minor changes

1. maybe summarize the abstract as one paragraph (delete empty line)
2. page 1 “deep neural version” → “deep net version”?
3. P.D.E → “P.D.E.” ?
4. $\mathbf{R}^2 \ni$ xxx $\in \mathbf{R}^2$ is strange; you don’t need double of this $\in$.
5. period in equation 1
6. page 3; the pendulum at time $t = x_t$; this equal is weird.
7. Informal theorem and formal theorem may want to have unified numbering or so; otherwise mentioning theorem 3.2 or so before introducing it is strange.
8. page 4 last paragraph is a bit hard to parse.
9. in Definition 2;  I would avoid using \& symbol just to mean “and” in math environment.
10. in Definition 2; the way f’ is defined is a bit strange; maybe it is ok but this is too intuitive definition that might cause misunderstanding.
11. in page 6; what do you mean by “… and facing bounded data”?
12. in page 8; the definition of $\mathcal{H}$ is a bit unclear (you could put the set as a condition in the set)
13. did you define $\rho,T$ in page 8?
14. $G$ and $\mathcal{G}$
15. in page 8 DeepONet (G) is confusing as it might mean that the DeepONet is a function taking G
16. bullet points in page 8; equal $=$ is used in a weird way.
17. there are may places where the parenthesis is not closed or overly closed; please check them.
18. in page 27; first inquality no $L$ needed.  Third inquality is equality?
19. the notation for uniform distribution is introduced before it is defined in page 28.
20. please define $|\mathcal{P}+B|$.
21. second line of Lemma I.2.  “function in we” → “function then we”?  Please check other typos as well.

**Strengths And Weaknesses:**

Although the techniques are simple, it is overall an interesting work.

Strength: The logical flow and the motivation are clear.   Simple yet interesting and important contributions that study generalization bounds on a specific instance of neural network architectures.

Major weakness:
The main concern is its structure and writing in some parts.
It seems that several claims and descriptions are somewhat scattered around the paper.  And some sentences are a bit unclear to me.
Also, mathematical expressions in some parts are unclear as well.

---

> ### Author Response · Authors · 2024-10-17
> **Responses to the Comments**
>
> We thank you for your careful reading of our draft and your detailed comments. We have implemented as many of your suggestions and below we give our detailed responses to the main queries.
>
> >1. There are several comparisons to existing work; please summarize them concisely in one section or so so that the relation can be clear.
>
> We have now created a consolidated Section-2 called ``Related Works'' where we have now reviewed in a single place the various previous literature on deep-learning generalization bounds.
>
> >2. It seems the authors separate meta summary of the work and mathematical setups, which looks good to me; however, if that is the case, meta summary part may need to be more intuitive so that there won’t be much overlap with the mathematical setup parts.
>
> We have now restructured the mathematical setup in Section 3, to not repeat inside it any part of the defining equations of a DeepONet, that already get introduced in the previous sections.
>
> >3. Experimental sections are not really well structured; please make it flow well and add sufficient descriptions to each figure. Also the texts in figures are a bit too small.
>
> We have changed the image quality to scalable vector graphic (svg) format which provides better resolution.
>
> >4. It says the complexity does not (explicitly) depend on width and depth; how does the constants C affect the complexity? It seems exponential?
>
> We suggest that the bounds should be read by first choosing the function space of which the Rademacher complexity is being computed and this function space of DeepONets, at any fixed depth, is being specified as that subset whose weights satisfy equation 4. Hence, the dependency of the Rademacher upperbound on all $\mathcal{C}$ terms should be read as a constant, and this way we conclude that the bound is not explicitly dependent on the width and hence the number of trainable parameters of the neural net. We note that this interpretation is consistent with the usual conventions of generalization-bound theory of neural nets - as reviewed in Section 2. Thus we can say that we have obtained size-independent generalization bounds - even for DeepONets and also while not making any assumptions on the target operator $\mathcal{G}$.
>
> >5. Definition 6 is a bit hard to parse.
>
> Kindly note that we have restructured Definition 6 for improved readability.
>
> >6. page 28 bottom; “since the supremum would be positive…” can you elaborate more on this?
>
> Kindly note that we have restructured the paragraphs and expanded on certain points to make the argument more readable for Lemma H.2.
>
> >7. $f$ and $f'$ are introduced; it seems $f'$ is used differently in different parts. (like in page 26) Can you unify them?
>
> Please note that in Lemma 8.1 the notation has been changed to not have any $f'$

---

> > ### Comment · Reviewer_mR3e · 2024-10-23
> > **Thank you for the response**
> >
> > Thank you for the response; the authors have addressed my concerns.

---

### Review · Reviewer_Vq8r · 2024-10-07

**Summary Of Contributions:**

This paper provides a theoretical and experimentally verified bounds on the complexity of the o.d.e. learning using a specific type of artificial neural networks called the DeepONets. This type of networks train simultaneously two networks in parallel one for the functions it self and one for the time step. The networks then combine their outputs to generate a value and the time index of the o.d.e. The main work in this paper is purely theoretical and the authors demonstrate that unlike in Lanthaler et. al showing that the generalization grows proportionally to the exponential growth of learning parameters, here the results demonstrate that the generalization is proportional to the capacity of the network. While these two seems to be independent the capacity of the network is a less precise measure especially in the complex methods and thus the result is interesting. This is achieved in particular when the network is trained using a specific type of loss, notably the Huber loss. The result is a bound obtained by Rademacher complexity that describes the ability of representing a function by a random encoder and in particular on the number of samples in the training set. As such the authors are able to reduce the problem to a learner independent approach.

**Audience:**

Yes

**Broader Impact Concerns:**

My only question is how is the Huber loss useful in the training of the o.d.e. That is is a standard approach or does it require specific set of preparatory steps to be applied?

**Claims And Evidence:**

Yes

**Requested Changes:**

Explain in more details the actual difference between Lanthaller and this work

**Strengths And Weaknesses:**

The strong side of the paper is the presentation of the result and the application of the Huber loss to training

My concern is the formulation of parameter independence. The Huber loss is directly proportional only to the accuracy of the sequence of itself and the Rademacher complexity models a function as a random system learner.  So doesn't this mean that actually the complexity represents the learnability or generalization in terms of the function complexity projected to the unitary encoder, the random encoding? Therefore I am not sure if both this work and Lanthaler work are not equivalent

---

> ### Author Response · Authors · 2024-10-17
> **Responses to the Comments**
>
> We thank you for your careful reading of our draft and your detailed comments. We have implemented as many of your suggestions and below we give our detailed responses to the main queries.
>
> >1. Rademacher complexity models a function as a random system learner. So doesn't this mean that actually the complexity represents the learnability or generalization in terms of the function complexity projected to the unitary encoder, the random encoding?
>
> In Theorem G.1 of our appendix, we recall the connection between Rademacher complexity and generalization error. Notably, interpreting this result does not rely on random weights, instead it provides a uniform upper bound over the entire class of loss functions parameterized by the weights allowed for. For a fixed loss function, the Rademacher complexity upperbounds the worst generalization gap in the allowed set of predictors - and intriguingly it does so by measuring how close anyone in the class gets to being able to to predict random binary noise as the labels.
>
> >2. Explain in more details the actual difference between Lanthaler and this work
>
> In Section 2.1, we discussed how our work relates to that of Lanthaler et al. (2022). We noted that none of the theorems presented in our work rely on their results, and the proof strategies are entirely independent. Theorem 5.3 in Lanthaler et al. (2022) is the only existing result on generalization error bounds for DeepONets. However, their bound has an explicit dependence on the total number of parameters in the DeepONet, denoted as $d_\theta$ in their notation.
>
> Furthermore, for usual implementations of DeepONets, where depths are typically small and the layers are wide, we observe that for a class of DeepONets with any fixed value of our complexity measure i.e. $\mathcal{C}\_{n,n-1}(\prod_{j=2}^{n-1}\mathcal{C}\_{-j,-j})$, the generalization error bound derived from our Rademacher complexity bound in Theorem 4.1 is smaller than the bound in Lanthaler et al. (2022), which scales with the total number of parameters.
>
> >3. My only question is how is the Huber loss useful in the training of the o.d.e. That is a standard approach or does it require specific set of preparatory steps to be applied?
>
> The only change required to use the Huber loss in neural networks is to replace the loss function, no additional modifications are needed.
>
> In addition to our previous experiments on the Burgers' PDE, we have now included experiments using the Huber loss on the Heat PDE in Section 4.3.2. Here too we demonstrate that the Rademacher bound under Huber loss exhibits a strong correlation with the generalization error across varying amounts of training data, for two distinct Huber delta values. In the newly added Appendix M we have shown that even beyond the ambit of our assumptions, DeepONet trained with Huber loss performs better than $l_2$ loss.

---

### Decision · Action_Editor_iEeD · 2024-11-17

**Recommendation:** Accept as is

**Comment:**

Given that the three reviewers unanimously agree on both the claims and audience questions, it seems a clear accept.

**Audience:**

There is a clear audience at TMLR.  It discusses generalization bounds for neural networks that solve PDEs.

**Claims And Evidence:**

There is unanimous agreement among the reviewers that the paper's claims are all supported with accurate, convincing and clear evidence.

---

> ### Author Response · Authors · 2024-11-27
> **Official Comment by Authors**
>
> We have submitted the de-anonymized camera-ready version and would like to thank everyone involved in the review process for their time. Please let us know if there is anything else needed from our side, before the displayed status of the paper changes to accepted.